# One-Shot Imitation Learning:
# A Pose Estimation Perspective

**Pietro Vitiello**[*]
The Robot Learning Lab
Imperial College London
`pv2017@ic.ac.uk`

**Kamil Dreczkowski**[*]
The Robot Learning Lab
Imperial College London
`krd115@ic.ac.uk`

**Edward Johns**
The Robot Learning Lab
Imperial College London
`e.johns@imperial.ac.uk`

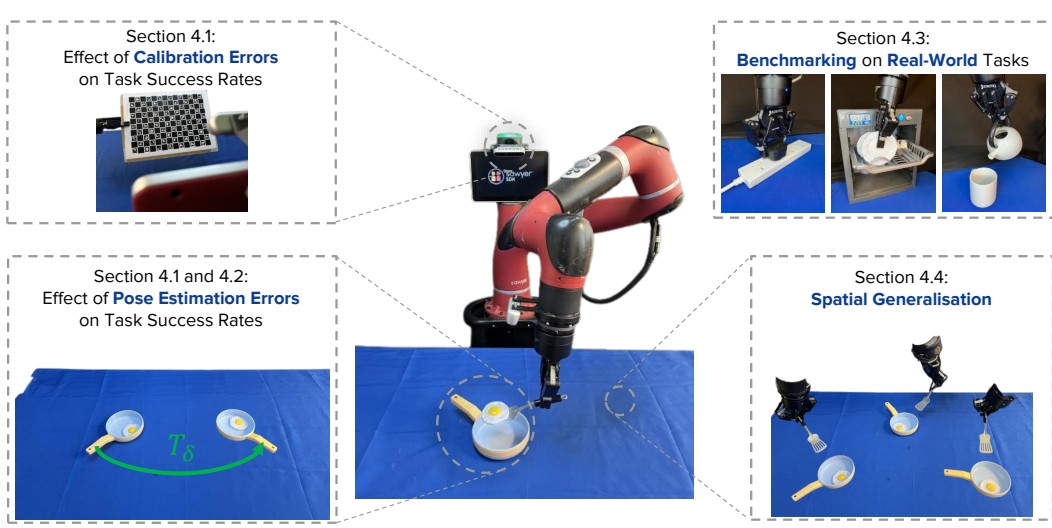

Figure 1: We model **one-shot imitation learning** as **trajectory transfer**, where we use **unseen object pose estimation** to adapt an end-effector trajectory from a single demonstration, to a new scene where the object is in a novel pose. In this paper, we are going to study this formulation through a series of four investigations shown in the above boxes.

**Abstract:** In this paper, we study imitation learning under the challenging setting of: (1) only a single demonstration, (2) no further data collection, and (3) no prior task or object knowledge. We show how, with these constraints, imitation learning can be formulated as a combination of trajectory transfer and unseen object pose estimation. To explore this idea, we provide an in-depth study on how state-of-the-art unseen object pose estimators perform for one-shot imitation learning on ten real-world tasks, and we take a deep dive into the effects that camera calibration, pose estimation error, and spatial generalisation have on task success rates. For videos, please visit www.robot-learning.uk/pose-estimation-perspective.

**Keywords:** One-Shot Imitation Learning, Unseen Object Pose Estimation, Robot Manipulation

## 1 Introduction

Imitation Learning (IL) can be a convenient and intuitive approach for teaching a robot how to perform a task. However, many of today's methods for learning vision-based policies require tens to hundreds of demonstrations per task [1, 2, 3, 4, 5, 6]. Whilst combining with reinforcement learning [7, 8, 9] or pre-training on similar tasks [10, 11, 12] can help, in this paper we take a look at **one-shot imitation learning**, where we assume: (1) only a single demonstration, (2) no further data collection following the demonstration, and (3) no prior task or object knowledge.

---

[*]Joint First Author Contribution

7th Conference on Robot Learning (CoRL 2023), Atlanta, USA.

With only a single demonstration and no prior knowledge about the task or the object(s) the robot is interacting with, the optimal imitation is one where the robot and object(s) are aligned in the same way as during the demonstration. For example, imitating a "scoop the egg" task (see Figure 1) could be achieved by aligning the spatula and the egg with the same sequence of relative poses as was provided during the demonstration.

But without any prior knowledge about the object(s), such as 3D object models, the reasoning required by the robot now distils down to an **unseen object pose estimation** problem: the robot must infer the relative pose between its current observation of the object(s) and its observation during the demonstration, in order to perform this trajectory transfer [13] (see Figure 2). Unseen object pose estimation is already a challenging field within the computer vision community [14, 15, 16, 17], and these challenges are further compounded in a robotics setting.

From this standpoint, we are the first to study the utility of unseen object pose estimation for trajectory transfer in the context of one-shot IL, and we reveal new insights into the characteristics of such a formulation and how to mitigate its challenges. We begin our study by analysing how camera calibration and pose estimation errors affect the success rates of ten diverse real-world manipulation tasks, such as inserting a plug into a socket or placing a plate in a dishwasher rack.

Following this, we estimate the pose estimation errors of eight different unseen object pose estimators in simulation, including one based on NOPE [18], a state-of-the-art unseen object orientation estimation method, and one based on ASpanFormer [19], a state-of-the-art correspondence estimation method. We then benchmark trajectory transfer using these eight unseen object pose estimators against DOME [20], a state-of-the-art one-shot IL method, on the same ten real-world tasks as mentioned above. Our results not only show that the unseen object pose estimation formulation of one-shot IL is capable of outperforming DOME by $22\%$ on average, but it is also applicable to a much wider range of tasks, including those for which a third-person perspective is necessary [21].

Finally, we evaluate the robustness of this formulation to changes in lighting conditions, and conclude our study by investigating how well it generalises spatially, as an object's pose differs to its pose during the demonstration.

## 2 Related Work

Whilst there are many methods that study imitation learning with multiple demonstrations per task [3, 22, 23], in this section, we set our paper within the context of existing one-shot IL methods.

**Trajectory Transfer**. Trajectory transfer refers to adapting a demonstrated trajectory to a new test scene. Previous work has considered how to warp a trajectory from the geometry during the demonstration to the geometry at test time [13], focusing on non-rigid registration for manipulating deformable objects. However, when relying on only a single demonstration, they displayed very local generalisation to changes in object poses, suggesting the need for multiple demonstrations in order to achieve greater spatial generalisation [13, 24, 25]. In contrast, we are the first to study unseen object pose estimation for trajectory transfer, which enables spatial generalisation from only a single demonstration.

**Methods that require further data collection**. Since one demonstration often does not provide sufficient information to satisfactorily learn a task, some methods rely on further data collection. For instance, Coarse-to-Fine approaches [26, 27] train a visual servoing policy by collecting data around the object in a self-supervised manner. On the other hand, FISH [9] fine-tunes a base policy learned with IL using reinforcement learning and interactions with the environment. While these approaches have their strengths, the additional environment interactions require time and sometimes human supervision. In contrast, modelling one-shot IL as unseen object pose estimation avoids the need for real-world data collection, hence enabling scalable learning.

**Methods that require prior knowledge**. Another way of compensating for the lack of demonstration data is to leverage prior task knowledge. For instance, many IL methods require access to object poses [28, 29, 30] or knowledge of the manipulated object categories [31], which is often impractical in everyday scenarios. Another approach that assumes prior knowledge for learning tasks from a single demonstration is meta-learning [10, 11, 12, 32, 33, 34, 35]. In this paradigm, a policy is pre-trained on a set of related tasks in order to infer actions for similar tasks from a single demonstration. However, the applicability of the learned policy is limited to tasks closely related to the meta-training dataset. Contrary to the meta-learning formulation of one-shot IL, we approach it as unseen object pose estimation, which assumes no prior knowledge and thus increases its generality.

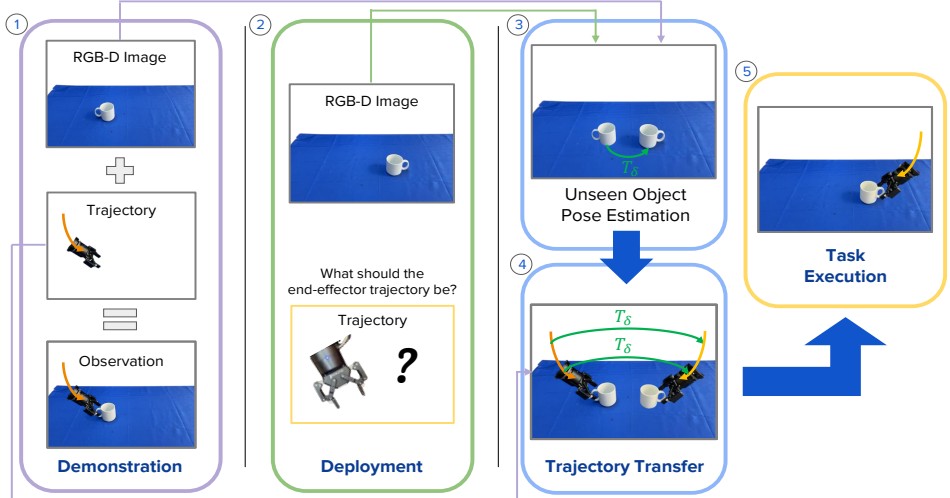

Figure 2: **Overview of our formulation for one-shot IL**: (1) The robot receives a demonstration as an RGB-D image and an end-effector trajectory. (2) At deployment, the robot sees the object in a new pose and must adapt the demonstrated trajectory accordingly. (3) To do so, the robot uses unseen object pose estimation to estimate the object transformation between demonstration and deployment. (4) It then applies this transformation to the demonstrated trajectory. (5) Ultimately, this aligns the end-effector with the same object-centric poses as experienced during the demonstration.

**Methods which do not require further training or prior knowledge**. DOME [20] and FlowControl [36] are one-shot IL algorithms that assume no prior knowledge. The effectiveness of these methods hinges on their reliance on a wrist-mounted camera, limiting their applicability to tasks where hand-centric observability is sufficient [21] and in which hand-held objects do not occlude the wrist-mounted camera. In contrast, the one-shot IL formulation explored in this paper applies to a much wider spectrum of tasks, including those for which a third-person perspective is necessary [21], such as the dishwasher task considered in our experiments (Section 4).

## 3   One-shot Imitation Learning for Robotic Manipulation

In this work, we study one-shot IL under the challenging setting when there is: (1) only a single demonstration, (2) no further data collection, and (3) no prior task or object knowledge. This is an appealing setting to aim for, since it encourages the design of a general and efficient method. In this section, we explore this from the perspective of object pose estimation. First, we provide a formulation of IL. Then, we model one-shot IL for manipulation as a trajectory transfer problem. Finally, we introduce unseen object pose estimation, which underpins the trajectory transfer problem.

Trajectory transfer, illustrated in Figure 2, involves the robot adapting a demonstrated end-effector (EEF) trajectory for deployment. This is done by estimating the relative object pose using unseen object pose estimation from a pair of RGB-D images captured at the beginning of the demonstration and deployment phases, allowing for spatial generalisation of the demonstrated trajectory.

### 3.1   Imitation Learning Formulation

We observe demonstrations as trajectories $\boldsymbol{\tau} = (\{\boldsymbol{x}_t\}_{t=1}^{T}, \boldsymbol{s})$, where $\boldsymbol{x}$ represents the state of the system, $T$ a finite horizon, and $\boldsymbol{s}$ the context vector. The state $\boldsymbol{x}$ encompasses various measurements relevant to the task and should include sufficient information for inferring optimal actions. In the context of robotic manipulation, the state could be the EEF and/or object(s) pose(s). Similarly, the context vector $\boldsymbol{s}$ captures various information regarding the conditions the demonstration was recorded under, and can assume different forms, ranging from an image of the task space captured before the demonstration to a task identifier in a multi-task setting.

Given a dataset of demonstrated trajectories $\mathcal{D} = \{\boldsymbol{\tau}_i\}_{i=1}^{N}$, IL aims to learn a policy $\pi^*$ that satisfies the objective:

$$\pi^* = \arg\min D(q(\boldsymbol{x}), p(\boldsymbol{x})), \tag{1}$$

where $q(\boldsymbol{x})$ and $p(\boldsymbol{x})$ are the distributions of states induced by the demonstrator and policy respectively, and $D(q, p)$ is a distance measure between $q$ and $p$.

## 3.2 Modelling One-Shot Imitation Learning as Trajectory Transfer

In this section, we model one-shot IL as trajectory transfer (see Figure 2), which we define as the process at test time of moving the EEF, or a grasped object, to the same set of relative poses, with respect to a target object, that it had during the demonstration. Let $R$ define the frame of the robot and $E_t$ that of the EEF at time step $t$. A homogeneous transformation matrix $\boldsymbol{T}_{AB}$ represents frame $B$ expressed in frame $A$. During demonstrations, the robot receives instructions through teleoperation or kinesthetic teaching, which are defined as:

$$\boldsymbol{\tau}^{Demo} = \left( \boldsymbol{X}_R^{Demo}, I^{Demo} \right), \tag{2}$$

and comprise of an RGB-D image $I^{Demo}$ of the task space captured before the demonstration using an external camera, and a sequence of EEF poses from the demonstration, $\boldsymbol{X}_R^{Demo} = \{\boldsymbol{T}_{RE_t}^{Demo}\}_{t=1}^T$, expressed in the robot frame.

With only a single demonstration and no prior knowledge about the object the robot is interacting with, the optimal imitation can be considered to be where the robot and object(s) are aligned in the same way as during the demonstration, throughout the task. For example, imitating grasping a mug (see Figure 2), could be achieved by aligning the EEF in such a way that the relative pose between the EEF and mug are the same as during the demonstration. Note that this also holds for more complex trajectories beyond grasping, such as insertion or twisting manoeuvres. Moreover, if the task involves a grasped object, assuming that the latter is fixed and rigidly attached to the gripper, aligning the EEF will also align the grasped object.

Now, consider Equation 1 in the context of manipulation, where the optimal policy $\pi^*$ should result in replicating the demonstrated task state, $\boldsymbol{x} = \boldsymbol{T}_{OE}$, where $O$ is the object frame, at every timestep during deployment. This demonstrated sequence of EEF poses, expressed in the object frame, is defined as $\boldsymbol{X}_O^{Demo} = \{\boldsymbol{T}_{OE_t}^{Demo}\}_{t=1}^T$, where

$$\boldsymbol{T}_{OE_t}^{Demo} = \boldsymbol{T}_{OR}^{Demo} \boldsymbol{T}_{RE_t}^{Demo} = \left( \boldsymbol{T}_{RO}^{Demo} \right)^{-1} \boldsymbol{T}_{RE_t}^{Demo}, \tag{3}$$

and $\boldsymbol{T}_{RO}^{Demo}$ is the unknown object pose during the demonstration. During deployment, for optimal imitation, $\pi^*$ should replicate $\boldsymbol{X}_O^{Demo}$ given a novel unknown object pose $\boldsymbol{T}_{RO}^{Test}$, i.e. we would like $\boldsymbol{T}_{OE}^{Test} = \boldsymbol{T}_{OE}^{Demo}$ during every timestep of the interaction. Hence, the sequence of EEF poses (expressed in the robot frame) that aligns with the demonstration can be defined as $\boldsymbol{X}_R^{Test} = \{\boldsymbol{T}_{RE_t}^{Test}\}_{t=1}^T$, where

$$\boldsymbol{T}_{RE_t}^{Test} = \boldsymbol{T}_{RO}^{Test} \boldsymbol{T}_{OE_t}^{Demo}. \tag{4}$$

Substituting Equation 3 into Equation 4 yields

$$\boldsymbol{T}_{RE_t}^{Test} = \boldsymbol{T}_{RO}^{Test} \boldsymbol{T}_{OR}^{Demo} \boldsymbol{T}_{RE_t}^{Demo} = {}^R\boldsymbol{T}_\delta \boldsymbol{T}_{RE_i}^{Demo}, \tag{5}$$

where we define

$$ {}^R\boldsymbol{T}_\delta = \boldsymbol{T}_{RO}^{Test} \boldsymbol{T}_{OR}^{Demo}, \tag{6}$$

which represents the transformation of the object between the demonstration and deployment scenes, where we use the superscript $R$ to indicate that ${}^R\boldsymbol{T}_\delta$ is expressed in the robot frame $R$.

This then leads to the crux of our investigation: the trajectory transfer problem, i.e. computing $\boldsymbol{X}_R^{Test}$ from $\boldsymbol{X}_R^{Demo}$, distils down to the problem of estimating the relative object pose between the demonstration and deployment scenes, given a single image from each. Once this pose is estimated, controlling the EEF to follow this trajectory can simply make use of inverse kinematics. And given that we assume no prior object knowledge, the challenge becomes one of one-shot unseen object pose estimation, an active field in the computer vision community [14, 15, 16, 17, 18].

## 3.3 One-shot Unseen Object Pose Estimation for Trajectory Transfer

One-shot unseen object pose estimation is concerned with estimating the relative pose of a novel object visible in two images. Formally, let $C$ denote the frame of reference of a camera, and consider one image $I^{Demo}$, taken when a novel object was at a pose $\boldsymbol{T}_{CO}^{Demo}$, and a second image $I^{Test}$, taken when that same object was at a pose $\boldsymbol{T}_{CO}^{Test}$. One-shot unseen object pose estimation aims to estimate the relative transformation between the two object poses, ${}^C\boldsymbol{T}_\delta$, that satisfies $\boldsymbol{T}_{CO}^{Test} = {}^C\boldsymbol{T}_\delta \boldsymbol{T}_{CO}^{Demo}$, where we use the superscript to indicate that ${}^C\boldsymbol{T}_\delta$ is expressed in the camera frame $C$. Rearranging this equation yields

$$ {}^C\boldsymbol{T}_\delta = \boldsymbol{T}_{CO}^{Test} \left( \boldsymbol{T}_{CO}^{Demo} \right)^{-1} = \boldsymbol{T}_{CO}^{Test} \boldsymbol{T}_{OC}^{Demo}. \tag{7}$$

Comparing Equations 6 and 7 reveals that $^{C}\boldsymbol{T}_{\delta}$ and $^{R}\boldsymbol{T}_{\delta}$ both represent the relative object pose, but are expressed in different frames of reference. In fact, after estimating $^{C}\boldsymbol{T}_{\delta}$ using one-shot unseen object pose estimation, we can find $^{R}\boldsymbol{T}_{\delta}$ from the following relationship derived in Appendix A:

$$^{R}\boldsymbol{T}_{\delta} = \boldsymbol{T}_{RC}\,^{C}\boldsymbol{T}_{\delta}\,(\boldsymbol{T}_{RC})^{-1} = \boldsymbol{T}_{RC}\,^{C}\boldsymbol{T}_{\delta}\boldsymbol{T}_{CR}, \tag{8}$$

where $\boldsymbol{T}_{RC}$ is the pose of the camera in the robot frame. Hence, the trajectory transfer problem can now be solved by using one-shot unseen object pose estimation to calculate the value of $^{C}\boldsymbol{T}_{\delta}$.

Examining Equation 8 reveals that there are two potential sources of error that could degrade the accuracy of $^{R}\boldsymbol{T}_{\delta}$ and compromise performance during deployment. The first source of error is the error in extrinsic camera calibration $\boldsymbol{T}_{RC}$, and the second is the error in unseen object pose estimation itself $^{C}\boldsymbol{T}_{\delta}$, both of which we discuss and study in the following sections.

## 4 Experiments

We now introduce ten representative everyday robotics tasks that span a broad range of complexities. As depicted in Figure 3, these tasks are: placing one bowl into another (**Bowls**), inserting a plug into a socket (**Plug**), grasping a mug by the handle (**Mug**), scooping an egg from a pan (**Egg**), inserting bread into a toaster (**Toaster**), inserting a plate into a specific slot in a dish rack (**Dishwasher**), inserting a cap into a bottle (**Cap**), pouring a marble from a kettle into a mug (**Tea**), grasping a can (**Can**), and placing a lid onto a pot (**Lid**).

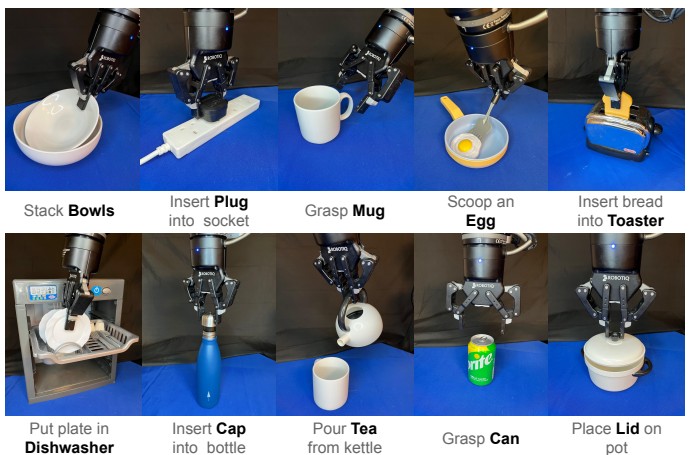

Figure 3: The 10 real-world tasks we use for evaluation.

We begin this section by studying the effect of calibration and pose estimation errors on the success rate for each of these tasks (Section 4.1). We then consider eight unseen object pose estimation methods and estimate their pose estimation errors in simulation (Section 4.2). And finally, we benchmark these pose estimation methods when used for trajectory transfer on the discussed real-world tasks (Section 4.3), we study the robustness to changes in lighting (Section 4.3) and distractors (Appendix G.2), and we examine the spatial generalisation capabilities of trajectory transfer (Section 4.4). For videos of our experiments, please visit our website.

### 4.1 Sensitivity Analysis of Task Success Rates to Calibration and Pose Estimation Errors

Correlating task success rates with calibration and pose estimation errors in the real world is challenging. To establish a relationship between these errors and task success rates, we begin by providing a single demonstration via kinesthetic teaching from a last-inch setting. We then measure the correlation between the task success rate and the starting EEF position error prior to imitating the demonstration (see Appendix F.2). Finally, we map starting EEF position errors to either calibration or pose estimation errors using an empirically defined mapping (see Appendix B).

Specifically, for each considered task, we reset the object position, add a position error to the starting EEF pose, replay the demonstration, and note if the task execution is successful. This is repeated 10 times for each position noise magnitude, with noise magnitudes starting from 0 mm and increasing in 2 mm increments until the success rate is 0% over three consecutive noise magnitudes. This resulted in a total of approximately 1,500 real-world trajectories in order to establish the relationship between EEF position errors and task success rates for the considered tasks.

Then, to empirically map calibration errors or pose estimation errors to these starting EEF position errors, only one potential source of error was considered at a time. For example, when mapping translation errors in calibration to starting EEF position errors, we assumed that rotation errors in calibration as well as rotation and translation errors in pose estimation are all zero, which isolates the effect of translation errors in calibration on task success rates.

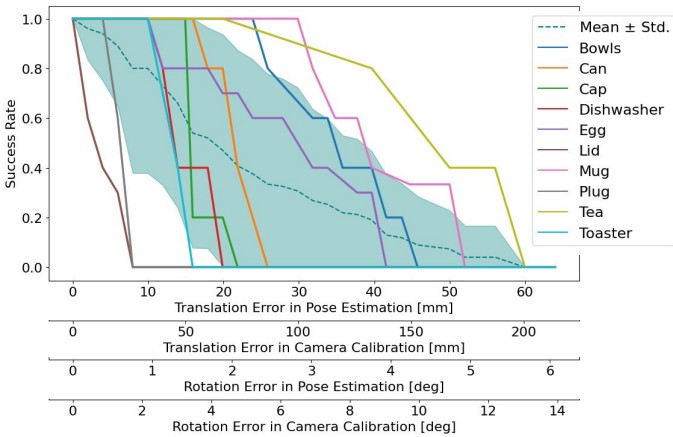

Figure 4: Correlation between error magnitudes in either calibration or pose estimation, and task success rates, assuming a distance of 80 cm between the camera and the task space.

The results for this experiment are shown in Figure 4, where each of the x-axes corresponds to a different type of error in calibration or pose estimation. The shape of the graph is identical for each error type, because there is a linear relationship between each of these errors and the error in starting EEF position. From these results, we can draw two main conclusions. Firstly, the task success rate for all tasks is more sensitive to errors in pose estimation than errors in camera calibration (for both translation and rotation errors), highlighting the importance of good pose estimation methods for this framework. Secondly, given typical performance of pose estimation and calibration methods, rotation errors are more problematic than position errors. For example, considering that rotation errors $< 4°$ are more probable than position errors $> 2$ cm with today's methods, we can see that a mere $\sim 4°$ error in calibration or $\sim 2°$ in pose estimation leads to an average success rate of $\sim 50\%$, whereas this same success rate would require a $\sim 7$ cm error in calibration or $\sim 2$ cm in pose estimation.

## 4.2 Pose Estimation Errors of One-Shot Unseen Object Pose Estimation Methods

We now consider eight different unseen object pose estimation methods and evaluate their pose estimation errors in simulation. To this end, we generate a simulated dataset consisting of 1100 image pairs of 55 different objects from Google Scan Object [37], using Blender [38] (see Appendix D.2).

The considered methods are: 1) **ICP**: We use the Open3D [39] implementation of point-to-point ICP [40]. ICP is given a total of 5 seconds to make each estimate, giving it enough time to try $50-150$ different initialisations. 2) **GMFlow**: We use GMFlow [41]

|            | Translation Error [cm] | Rotation Error [deg] |
|------------|------------------------|----------------------|
| Class.     | **5.9 ± 11.2**         | **4.3 ± 10.1**       |
| ASpan. (FT)| 6.0 ± 13.0             | 6.1 ± 13.4           |
| Reg.       | 9.8 ± 15.3             | 9.4 ± 15.3           |
| DINO       | 11.3 ± 18.2            | 11.6 ± 18.9          |
| ASpan.     | 11.5 ± 17.4            | 11.7 ± 18.8          |
| NOPE       | 18.8 ± 17.4            | 18.4 ± 17.2          |
| ICP        | 14.3 ± 30.8            | 14.4 ± 32.4          |
| GMFlow     | 28.4 ± 24.6            | 27.6 ± 24.5          |

Table 1: Simulation pose estimation errors and standard deviations for eight different methods.

to estimate correspondences between two RGB images and solve for the relative object pose $^C T_\delta$ with Singular Value Decomposition (SVD) [42] using the depth image. 3) **DINO**: We use DINO [43] to extract descriptors for pixels in the two RGB images, and use the SciPy [44] implementation of the Hungarian algorithm to establish correspondences. Again, the relative pose $^C T_\delta$ is obtained using SVD. 4) **ASpan.**: We use the pre-trained ASpanFormer [19] to establish correspondences between two RGB images and estimate $^C T_\delta$ using SVD. 5) **ASpan. (FT)**: The ASpan. baseline, with model weights fine-tuned on a custom object-centric dataset generated in Blender using ShapeNet and Google Scan Objects [37]. 6) **NOPE**: We use the pre-trained NOPE [18] model to estimate the relative object rotation from two images, and a heuristic that centres two partial point clouds to predict the relative translation. 7) **Reg.**: We train an object-agnostic PointNet++ to regress relative object orientations around the world's vertical axis from two coloured point clouds, using data generated in simulation and domain randomisation. We solve for the relative translation using a heuristic that centres two partial point clouds. 8) **Class.**: This is equivalent to the Reg. baseline with the exception that PointNet++ is trained to classify the relative object orientation.

We also experimented with predicting the relative object translation from pairs of RGB-D images for the NOPE, Reg. and the Class. baselines. However, we found that a simple heuristic that centres partial point clouds for translation prediction had a similar performance, and thus used this during inference. We refer the reader to Appendix C for a more detailed description of all of these methods.

| | Plug | Pot | Toaster | Dishwasher | Mug | Egg | Bottle | Tea | Bowls | Can | Mean |
|---|---|---|---|---|---|---|---|---|---|---|---|
| TT (GMFlow) | 10 | 40 | 0 | 20 | 40 | 20 | 20 | 20 | 60 | 60 | 29 |
| TT (ASpanFormer (FT)) | 0 | 10 | 10 | 50 | 60 | 50 | 50 | 30 | 30 | 50 | 34 |
| TT (ASpanFormer) | 0 | 10 | 0 | 20 | 50 | 50 | 50 | 40 | 60 | 70 | 35 |
| TT (DINO) | 0 | 20 | 10 | 30 | 50 | 60 | 40 | 40 | 80 | 70 | 40 |
| TT (NOPE) | 0 | 10 | 0 | 50 | 0 | 70 | 90 | **100** | 70 | **100** | 49 |
| DOME | 0 | 10 | 80 | 0 | **100** | 70 | 40 | 90 | 70 | **100** | 56 |
| TT (ICP) | 10 | **70** | 80 | 40 | 60 | 80 | **100** | **100** | **100** | **100** | 74 |
| TT (Class.) | **20** | 10 | **90** | 70 | **100** | **90** | **100** | **100** | 80 | **100** | 76 |
| TT (Reg.) | **20** | 30 | **90** | 70 | **100** | **90** | **100** | **100** | 80 | **100** | 78 |
| Mean | 6.7 | 23.3 | 40 | 46.7 | 62.2 | 64.4 | 65.6 | 68.9 | 70 | 83.3 | |

Table 2: Real-world success rates (%), from ten trials for each combination of method and task. TT (Trajectory Transfer) is used to distinguish all the previously discussed baselines from DOME [20].

The results for this experiment are shown in Table 1 (see Appendix E for an error definition and a discussion of these results). Although directly comparing these results to Figure 4 suggests that none of these baselines would be suitable for learning the considered tasks, in practice we found that translation and rotation errors in pose estimates are often coupled and partially cancel each other out, while Figure 4 only considers isolated errors. These observations are further reinforced by the strong performance we found with these baselines in our real-world experiments.

## 4.3 Real-World Evaluation

We now investigate if the trajectory transfer formulation of one-shot IL can learn real-world, everyday tasks, of varying tolerances.

**Implementation Details** For trajectory transfer, we use a given unseen object pose estimator to estimate $^{C}T_{\delta}$, and Equations 8 and 5 alongside inverse kinematics to align the robot with the first state of the demonstration. From this state, we align the full robot trajectory with the demonstrated trajectory, following Appendix F.2. In order to isolate the object of interest from the background, we segment it from the RGB-D image captured before the demonstration and deployment using a combination of OWL-ViT [45] and SAM [46]. Both segmented RGB-D images are subsequently downsampled to ensure compatibility with a given method. See Appendix F for further details.

**Experimental Procedure** We conduct experiments using a 7-DoF Sawyer robot operating at 30 Hz. The robot is equipped with a head-mounted camera capturing 640-by-480 RGB-D images. The task space is defined as a $30 \times 75$ cm region on a table in front of the robot, which is further divided into 10 quadrants measuring $15 \times 15$ cm each. During the demonstration phase, all objects are positioned in approximately the same location near the middle of the task space (see Figure 5), and a single demonstration is provided for each task via kinesthetic teaching from a last-inch setting. In the testing phase, the object is randomly placed within each quadrant with a random orientation difference of up to $\pm 45°$ relative to the demonstration orientation. We test each method on a single object pose within each of the quadrants, resulting in 10 evaluations per method.

**Results.** The results for this experiment are shown in Table 2, with tasks ordered by mean success rate across methods and methods ordered by mean success rate across tasks. These results also include a comparison against DOME [20], a state-of-the-art one-shot IL method. We observe that for DOME, the majority of failure cases are caused by its inaccurate segmentation of target objects. Its poor performance on the dishwasher task is attributed to the fact that the demonstration had to be started from further away, as DOME requires the object to be fully visible from a wrist-mounted camera. As a result, DOME was beaten on average by three of the baselines.

The Reg. and Class. baselines had the best performance on average, likely due to the fact that their training data was tailored to object manipulation (see Appendix D.1). ICP's performance was affected by the partial nature of the point clouds, which sometimes caused it to converge to local optimums. NOPE found itself out of its training distribution. Being trained on images with the object in the centre, NOPE can confuse a relative translation for a rotation when an object is displaced from the image centre. DINO uses semantic descriptors, which cause keypoints to be locally similar, translating into matches that are coarse and not precise. ASpanFormer was trained on images of entire scenes with many objects, hence expecting scenes rich with features. Therefore, predicting correspondences for a single segmented object causes this method to perform poorly. Meanwhile, we note that the fine-tuned ASpanFormer's performance drops significantly more with the sim-to-real gap than that of the Reg. and Class. methods. Lastly, GMFlow was found to poorly estimate rotations as the predicted flow tended to be smooth and consistent across pixels.

**Robustness to Changes in Lighting Conditions**. We now focus on trajectory transfer using regression, the best-performing method in our real-world experiments, and analyse its robustness to changes in lighting conditions. To this end, we rerun the real-world experiment for this method while additionally randomising the position, luminosity, and colour temperature of an external LED light source before each rollout. The results from this experiment indicate that trajectory transfer using regression remains strong, with an average decrease in performance of only $8\%$ when the lighting conditions are randomised significantly between the demonstration and test scene. We attribute this strong performance to the fact that the dataset used to train this baseline randomises lighting conditions between the two input images as part of domain randomisation. For full details regarding this experiment and its results, we refer the reader to Appendix G.1.

### 4.4 Spatial Generalisation

Another insight that emerged from the real-world experiments is the impact of the relative object pose between the demonstration and deployment on the average performance of trajectory transfer. When we aggregate the success rates across all baselines, tasks and poses within each of

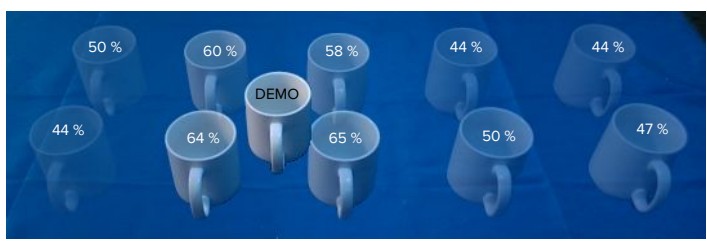

Figure 5: The correlation between success rate and displacement from the place where the demonstration was given.

the quadrants, we notice a decline in the success rate of trajectory transfer as the object pose deviates from the demonstration pose. In Figure 5, we display a mug at the approximate location where all objects were placed during demonstrations (labelled as DEMO), as well as a mug at the centre of each of the quadrants. The opacity of the mugs located in the different quadrants is proportional to the average success rate for those quadrants, which is also displayed in white text. Note that whilst in this figure the orientation of the mug is fixed, experiments did randomise the orientations.

The cause of this behaviour lies in the camera perspective. Specifically, even when kept at a fixed orientation, simply changing the position of an object will result in changes to its visual appearance. Moreover, contrary to the effect of errors in camera calibration (see Appendix B.1), the changes in the visual appearance lessen as the camera is placed further away from the task space. These insights might seem intuitive, but for this same reason, they could be easily overlooked by researchers in the field. As a result, for optimal spatial generalisation, we recommend providing demonstrations at the centre of the task space, as this minimises the variations in the object appearance when the object's pose deviates from the demonstration pose.

## 5  Discussion and Limitations

By formulating one-shot IL using unseen object pose estimation, we are able to learn new tasks without prior knowledge, from a single demonstration and no further data collection. We demonstrate this from a theoretical perspective and show its potential when applied to real-world tasks.

One limitation of this method is that we do not address generalisation to intra-class instances. Using semantic visual correspondences [47] is a promising future direction here. Another limitation is the reliance on camera calibration. However, our analysis of calibration errors and real-world experiments do indicate good performance given typical calibration errors.

Although the proposed method has demonstrated to be very versatile in the types of tasks it can learn, in our setup it required a static scene. This is because the robot arm often occludes the task space given the head-mounted camera on our Sawyer robot, making it not possible to continuously estimate the object pose during deployment. However, this is a limitation of the hardware setup and not a fundamental limitation of the method. By optimising the camera placement for minimum occlusions, trajectory transfer could be deployed in a closed-loop and in dynamic scenes.

Finally, the current formulation is unsuitable for tasks that depend on the relative pose between two objects, where neither of them is rigidly attached to the EEF. For instance, pushing an object close to another cannot rely on the rigid transfer of the trajectory, because the latter needs to be adapted according to the relative pose of the two objects. However, such tasks are fundamentally ambiguous with only a single demonstration, and multiple demonstrations would be required.

**Acknowledgments**

We would like to thank all reviewers for their thorough and insightful feedback, which had a significant impact on our paper.

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

# Appendix A Changing the Coordinate Frame of $T_\delta$

In the main paper, we state that the equation below changes the frame of reference of $T_\delta$ from the camera's frame $C$ to the robot's frame $R$:

$$^R T_\delta = T_{RC}{}^C T_\delta T_{CR} = T_{RC}{}^C T_\delta \left(T_{RC}\right)^{-1}, \tag{9}$$

where $T_{RC}$ is the pose of the camera in the robot frame. To derive this relationship, we begin by referring to the definition of $^C T_\delta$ presented in Equation 7 of the main paper:

$$^C T_\delta = T_{CO}^{Test} T_{OC}^{Demo} = T_{CO}^{Test} \left(T_{CO}^{Demo}\right)^{-1}.$$

Rearranging this equation yields:

$$T_{CO}^{Test} = {}^C T_\delta T_{CO}^{Demo}. \tag{10}$$

Additionally, we know that

$$T_{RO}^{Test} = T_{RC} T_{CO}^{Test}. \tag{11}$$

By substituting Equation 10 into Equation 11, we obtain:

$$\begin{aligned} T_{RO}^{Test} &= T_{RC} T_{CO}^{Test} \\ &= T_{RC}{}^C T_\delta T_{CO}^{Demo} \end{aligned} \tag{12}$$

However,

$$T_{CO}^{Demo} = T_{CR} T_{RO}^{Demo} = \left(T_{RC}\right)^{-1} T_{RO}^{Demo}.$$

By substitute this relationship into Equation 12 we obtain

$$T_{RO}^{Test} = T_{RC}{}^C T_\delta T_{CR} T_{RO}^{Demo},$$

which can be rearranged to obtain

$$T_{RC}{}^C T_\delta T_{CR} = T_{RO}^{Test} T_{OR}^{Demo} = T_{RO}^{Test} \left(T_{RO}^{Demo}\right)^{-1}.$$

Since the right-hand side of this equation is consistent with the definition of $^R T_\delta$ presented in Equation 6 of the main paper, we conclude that

$$^R T_\delta = T_{RC}{}^C T_\delta T_{CR}.$$

# Appendix B Mapping End-effector Errors to Calibration and Pose Estimation Errors

In Section 4.1 of the main paper, we detail our experimental procedure to derive the correlation between task success rates and starting end-effector position errors prior to replaying a last-inch demonstration. We then mention that we map these end-effector position errors to the corresponding calibration or pose estimation errors. In this section, we describe our procedure for deriving the empirical mapping between end-effector position errors and the corresponding calibration (Appendix B.1) and pose estimation errors (Appendix B.2).

## B.1 Errors in Camera Calibration

We derive the empirical relationship between end-effector position errors and camera calibration errors using an experiment in which we repeatedly (1) sample a starting end-effector pose for a demonstration, $T_{RE}^{Demo}$, (2) sample a relative object pose between the demonstration and deployment, $^C T_\delta$, and (3) compute the accuracy with which we can calculate the corresponding starting end-effector pose for deployment, $T_{RE}^{Test}$, using trajectory transfer, after injecting controlled amounts of noise to the camera calibration matrix $T_{RC}$.

**Experimental Procedure** We begin by calibrating a head-mounted camera to a Sawyer robot in the real world, obtaining an estimate of the camera pose:

$$T_{RC} = \left[R_{RC} | t_{RC}\right],$$

where $R_{RC} \in SO(3)$ is a rotation matrix and $t_{RC} \in \mathbb{R}^3$ is a translation vector. We then sample a random end-effector pose $T_{RE}^{Demo}$, which can be interpreted as the end-effector pose at the beginning

of a demonstration. Further, we sample a random relative object movement expressed in the camera frame,

$$^C\boldsymbol{T}_\delta = \begin{bmatrix} ^C\boldsymbol{R}_\delta | ^C\boldsymbol{t}_\delta \end{bmatrix},$$

where $^C\boldsymbol{R}_\delta \in SO(3)$ is a rotation matrix obtained from a randomly sampled rotation vector with a rotation magnitude sampled from the interval $[0, 45]°$, and $^C\boldsymbol{t}_\delta \in \mathbb{R}^3$ is a randomly sampled translation vector with a magnitude sampled from the interval $[0, 0.4]$m. The interval $[0, 45]°$ was chosen for consistency with our real-world experiments, while the maximum magnitude of the translation is equal to half of the longest side of our workspace, and corresponds to the maximum translation that is allowable between a demonstration and deployment when the demonstration is given with the object at the centre of the workspace.

After sampling the hypothetical end-effector pose $\boldsymbol{T}_{RE}^{Demo}$ and the relative object pose $^C\boldsymbol{T}_\delta$, we use them together with the camera extrinsic matrix $\boldsymbol{T}_{RC}$ to calculate the desired end-effector pose at test time,

$$\boldsymbol{T}_{RE}^{Test} = \begin{bmatrix} \boldsymbol{R}_{RE}^{Test} | \boldsymbol{t}_{RE}^{Test} \end{bmatrix},$$

using trajectory transfer (see Equations 5 and 8 in the main paper).

Once we compute the desired end-effector pose via trajectory transfer, we either perturb the rotation matrix of the camera calibration matrix, resulting in

$$\bar{\boldsymbol{T}}_{RC} = \begin{bmatrix} \boldsymbol{R}_\epsilon \boldsymbol{R}_{RC} | \boldsymbol{t}_{RC} \end{bmatrix},$$

where $\boldsymbol{R}_\epsilon \in SO(3)$ is a rotation matrix obtained from a randomly sampled rotation vector with a predetermined rotation magnitude, or perturb the translation vector of the camera calibration matrix, resulting in

$$\bar{\boldsymbol{T}}_{RC} = \begin{bmatrix} \boldsymbol{R}_{RC} | \boldsymbol{t}_{RC} + \boldsymbol{t}_\epsilon \end{bmatrix},$$

where $\boldsymbol{t}_\epsilon \in \mathbb{R}^3$ is a randomly sampled vector with a predetermined magnitude.

After perturbing the camera pose, we estimate the desired end-effector pose,

$$\bar{\boldsymbol{T}}_{RE}^{Test} = \begin{bmatrix} \bar{\boldsymbol{R}}_{RE}^{Test} | \bar{\boldsymbol{t}}_{RE}^{Test} \end{bmatrix},$$

using trajectory transfer and the noisy camera calibration matrix $\bar{\boldsymbol{T}}_{RC}$. Finally, we calculate the error between the ground truth and estimated end-effector pose, $\boldsymbol{T}_{RE}^{Test}$ and $\bar{\boldsymbol{T}}_{RE}^{Test}$, using the following equations:

$$t_{\text{error}} = \left\Vert \boldsymbol{t}_{RE}^{Test} - \bar{\boldsymbol{t}}_{RE}^{Test} \right\Vert_2 \qquad\qquad R_{\text{error}} = \left\Vert \log\left( \boldsymbol{R}_{RE}^{Test} \left( \bar{\boldsymbol{R}}_{RE}^{Test} \right)^T \right) \right\Vert_2$$

We repeat this procedure for 1000 different randomly sampled relative object poses $^C\boldsymbol{T}_\delta$, for hypothetical end-effector poses $\boldsymbol{T}_{RE}^{Demo}$ with end-effector-to-camera distances ranging from 0.2m to 1.2m, increasing in increments of 0.1m.

We present the results from this experiment in Figure 6. The first two graphs focus on translation errors in camera calibration and their impact on trajectory transfer. The bottom two graphs examine rotation errors in camera calibration and their influence on trajectory transfer.

**Interesting Findings** The top graph of Figure 6 reveals that translation errors in camera calibration lead to proportional errors in trajectory transfer. However, it is important to note that the magnitude of the errors in the starting end-effector positions are smaller compared to the magnitude of the calibration errors. Additionally, from the second graph we find that translation errors in camera calibration do not affect rotation errors in trajectory transfer, which aligns with our expectations.

Moving on to rotation errors in camera calibration (third graph in Figure 6), we notice that the translation error in trajectory transfer depends not only on the rotation error but also on the distance between the end-effector and the camera. This relationship is logical since rotations occur around the camera frame, and the resulting translation induced by an error in rotation is proportional to the distance from the frame of rotation. *Hence, a camera should be placed as close as possible to the robot's workspace to attenuate the effects of rotation errors in camera calibration on trajectory transfer.* Furthermore, we observe that rotation errors in camera calibration (last graph in Figure 6) correspond to proportional rotation errors in trajectory transfer, although the latter are consistently smaller in magnitude.

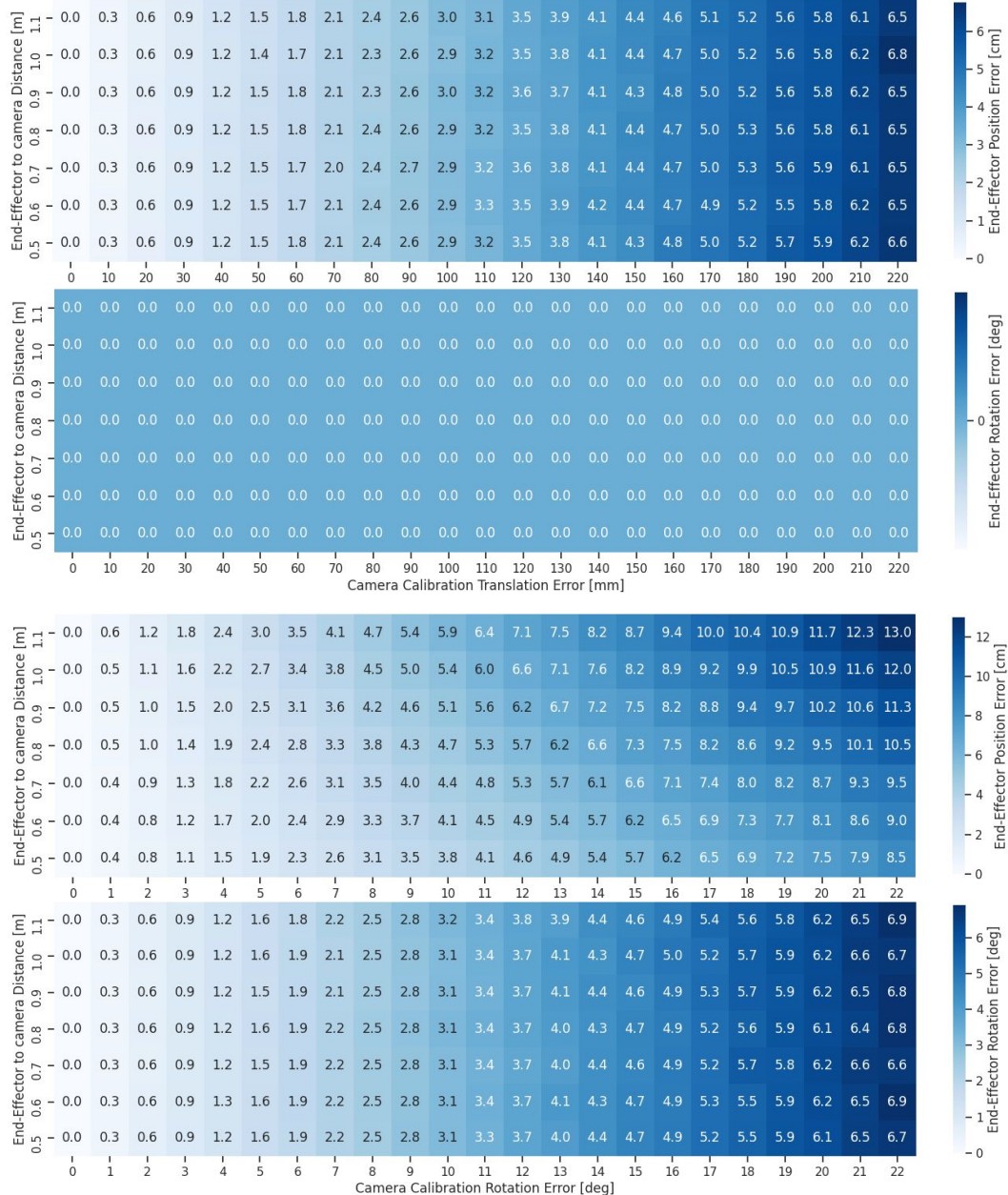

Figure 6: The relationship between the end-effector-to-camera distance, the magnitude of the error in camera calibration, and the error in trajectory transfer (i.e. the error in the calculated starting pose of the end-effector prior to replaying a demonstration during deployment).

**Mapping starting end-effector errors to calibration errors** To map starting end-effector position errors to translation errors in extrinsic calibration, we fit a first-order bivariate spline\* to map end-effector-to-camera distances and end-effector position errors to corresponding camera calibration translation errors (topmost graph of Figure 6). Similarly, to map starting end-effector position errors to rotation errors in camera calibration, we fit a first-order bivariate spline to map end-effector-to-camera distances and end-effector position errors to corresponding camera calibration rotation errors (third graph of Figure 6).

## B.2 Errors in Pose Estimation

The empirical relationship between end-effector position errors and pose estimation errors is derived using a very similar experimental procedure to that outlined above. However, instead of injecting controlled amounts of noise to the camera calibration matrix $\boldsymbol{T}_{RC}$, we now instead inject noise to the relative object pose $^{C}\boldsymbol{T}_{\delta}$.

**Experimental Procedure** Just like in the previous experiment, we begin by calibrating a head-mounted camera to a Sawyer robot in the real world, obtaining an estimate of the camera pose $\boldsymbol{T}_{RC}$. We then sample a random starting end-effector pose for a demonstration, $\boldsymbol{T}_{RE}^{Demo}$, and a relative object movement expressed in the camera frame, $^{C}\boldsymbol{T}_{\delta}$, using the same procedure as in the previous experiment. After sampling the starting end-effector pose $\boldsymbol{T}_{RE}^{Demo}$ and relative object pose $^{C}\boldsymbol{T}_{\delta}$, we use them and the camera extrinsic matrix $\boldsymbol{T}_{RC}$ to calculate the desired end-effector pose during deployment, $\boldsymbol{T}_{RE}^{Test}$, using trajectory transfer (see Equation 5 and 8 in the main paper).

Once we compute the desired end-effector pose via trajectory transfer, we either perturb the rotation matrix of the relative object pose $^{C}\boldsymbol{T}_{\delta}$, resulting in

$$^{C}\bar{\boldsymbol{T}}_{\delta} = \left[ \boldsymbol{R}_{\epsilon}{}^{C}\boldsymbol{R}_{\delta} | ^{C}\boldsymbol{t}_{\delta} \right],$$

where $\boldsymbol{R}_{\epsilon} \in SO(3)$ is a rotation matrix obtained from a randomly sampled rotation vector with a predetermined rotation magnitude, or perturb the translation vector of the relative object pose, resulting in

$$^{C}\bar{\boldsymbol{T}}_{\delta} = \left[ ^{C}\boldsymbol{R}_{\delta} | ^{C}\boldsymbol{t}_{\delta} + \boldsymbol{t}_{\epsilon} \right]$$

where $\boldsymbol{t}_{\epsilon} \in \mathbb{R}^{3}$ is a randomly sampled vector with a fixed magnitude.

We then estimate the desired end-effector pose,

$$\bar{\boldsymbol{T}}_{RE}^{Test} = \left[ \bar{\boldsymbol{R}}_{RE}^{Test} | \bar{\boldsymbol{t}}_{RE}^{Test} \right],$$

using trajectory transfer and the noisy relative object pose $^{C}\bar{\boldsymbol{T}}_{\delta}$, and calculate the error between the ground truth and estimated end-effector poses, $\boldsymbol{T}_{RE}^{Test}$ and $\bar{\boldsymbol{T}}_{RE}^{Test}$, using the same procedure as in the previous experiment. We repeat this for 1000 different randomly sampled relative object poses $^{C}\boldsymbol{T}_{\delta}$, for hypothetical demonstrated end-effector poses $\boldsymbol{T}_{RE}^{Demo}$ with end-effector-to-camera distances ranging from 0.2m to 1.2m in increments of 0.1m.

We present the results from this experiment in Figure 7. Just like in Figure 6, the first two graphs focus on translation errors in pose estimation and their impact on trajectory transfer, and the bottom two graphs focus on rotation errors in pose estimation and their influence on trajectory transfer.

**Interesting Findings** The top graph of Figure 7 reveals that translation errors in pose estimates lead to equal errors in trajectory transfer. Additionally, from the second graph of Figure 7 we find that translation errors in pose estimates do not affect rotation errors in trajectory transfer, which aligns with our expectations.

Moving on to rotation errors in pose estimation (third graph of Figure 7), we notice that the translation error in trajectory transfer depends not only on the rotation error but also on the distance between the end-effector and the camera. This relationship is expected since rotations occur around the camera frame, and the resulting translation induced by an error in rotation is proportional to the distance from the frame of rotation. Furthermore, we observe that rotation errors in pose estimates equal rotation errors in trajectory transfer (bottom graph of Figure 7). Finally, we note that the *errors in trajectory transfer induced by errors in pose estimation are far greater in magnitude than those induced by errors in calibration*.

---

\*We have experimented with fitting higher order bivariate splines. However, we have found the first-order spline to result in the lowest root mean squared error on a validation set.

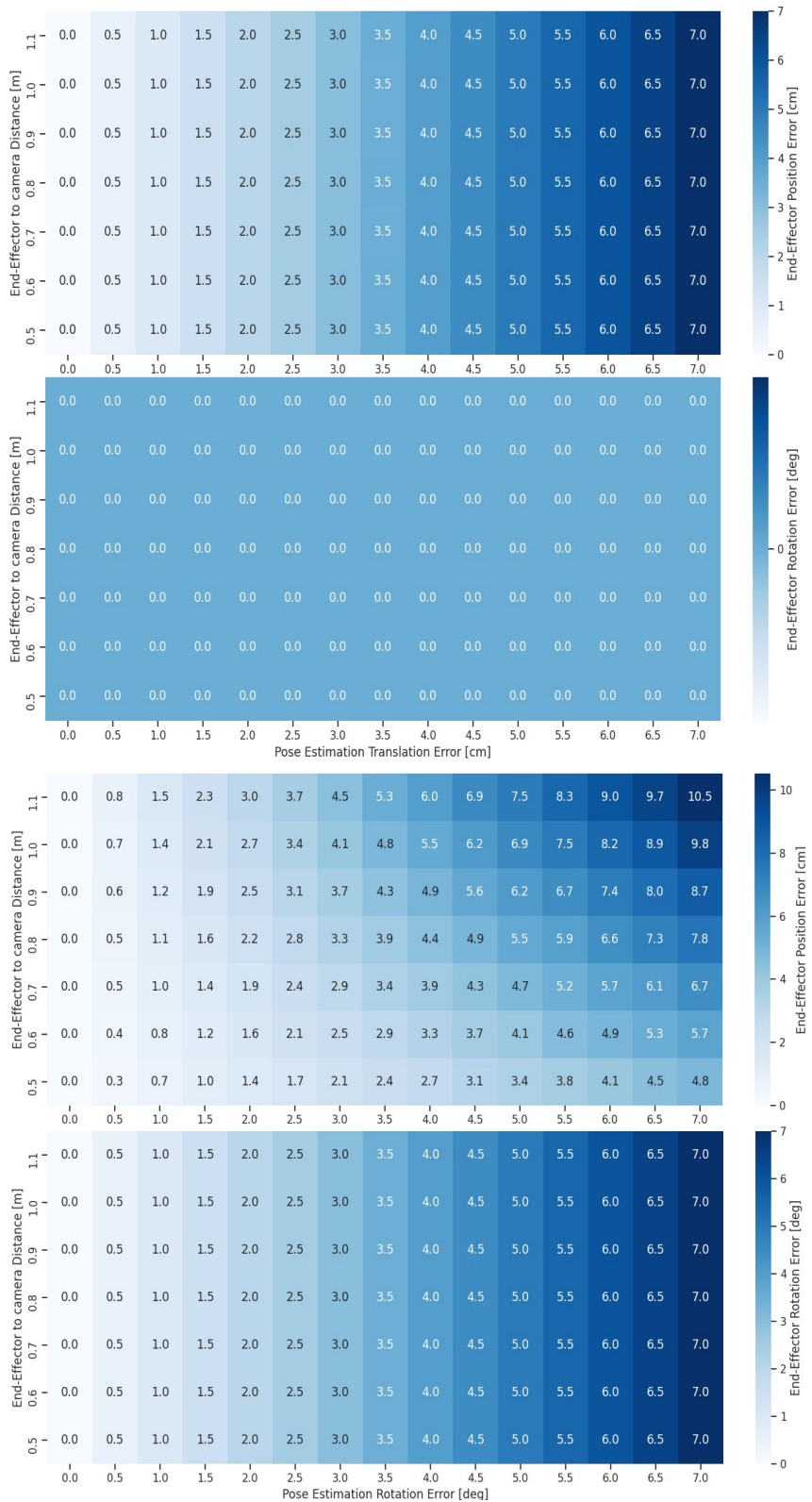

Figure 7: The relationship between the end-effector-to-camera distance, the magnitude of the error in pose estimation, and the error in trajectory transfer (i.e. the error in the calculated starting pose of the end-effector prior to replaying a demonstration during deployment).

**Mapping starting end-effector errors to pose estimation errors** To map starting end-effector position errors to translation errors in pose estimation, we fit a first-order bivariate spline* to map end-effector-to-camera distances and end-effector position errors to corresponding pose estimation translation errors (topmost graph of Figure 7). Similarly, to map starting end-effector position errors to rotation errors in pose estimation, we fit a first-order bivariate spline to map end-effector-to-camera distances and end-effector position errors to corresponding pose estimation rotation errors (third graph of Figure 7).

## Appendix C  Unseen Object Pose Estimation Baselines

### C.1  Iterative Closest Point

We use the Open3D [39] implementation of point-to-point ICP [40] to directly estimate $^C\boldsymbol{T}_\delta$. We set the maximum distance between correspondences to 10cm, the maximum number of ICP iterations to 10, and allow ICP to try as many random initialisations as possible within 5 seconds. This typically resulted in ICP trying approximately 100 random initialisations. We have tried increasing the maximum number of ICP iterations, but observed that it is better to try more random initialisations than to have more ICP iterations per initialisation.

To initialise ICP, we first sample a random rotation around the z-axis in the robot frame, and map this rotation to the camera frame using the camera extrinsic matrix. Sampling rotations in this way exploits the *prior knowledge that objects are translated in 3DoF while being rotated only around the robot's z-axis between the demonstration and deployment*, which is the case for the majority of manipulation tasks. To obtain the initialisation for translation, we first rotate the first partial point cloud using the sampled rotation, and then centre the two partial point clouds. Finally, we add Gaussian noise to the translation component with a standard deviation equal to 1cm.

### C.2  Correspondence-Estimation-Based Methods

We explore four different methods for estimating correspondences:

**DINO**: We use Deep ViT features [43] to establish correspondences between two RGB images.

**GMFlow**: We use the pre-trained GMFlow [41] model to predict optical flow, which is used to establish correspondences between two RGB images.

**ASpanFormer**: We use the pre-trained ASpanFormer [19] model to directly predict correspondences between two RGB images.

**ASpanFormer (FT)**: We fine-tune the pre-trained ASpanFormer model using an object-centric dataset generated in simulation (see Appendix D.1), and use it to directly predict correspondences.

After establishing correspondences using any of these methods, we apply a filtering step to remove outliers. To accomplish this, we leverage the Universal Sample Consensus (USAC) algorithm [48], which is an extension and generalisation of the Random Sample Consensus (RANSAC) algorithm [49]. Specifically, we utilise the OpenCV [50] implementation of USAC and set the RANSAC reprojection threshold to 5 pixels.

Once correspondences are established and outliers have been removed, all methods rely on Singular Value Decomposition (SVD) [42] to predict the relative object pose $^C\boldsymbol{T}_\delta$ using correspondences and their depth values.

### C.2.1  DINO

Our implementation of the DINO correspondence estimator begins by cropping segmented RGB images around their segmentation masks, and resizing them so that their longest side measures 224 pixels, while maintaining their original aspect ratio. Next, we extract DINO features from both segmented image crops using the pre-trained *dino_vit8* model [43] with a stride of 4. To establish correspondences, we compute the cosine similarity between the descriptors of all patches and employ the Hungarian Algorithm [51]. Once correspondences are established, we discard all correspondences with a cosine similarity lower than 0.1.

---

*We attempted fitting higher order splines but found the first-order spline to result in the lowest root mean squared error on a validation set.

### C.2.2 GMFlow

Our implementation of the GMFlow correspondence estimator begins by cropping the two (non-segmented) RGB images around their segmentation masks, and resizing them so that the width of the wider image measures 128 pixels, while preserving the original aspect ratio of both images. Then, the pre-trained GMFlow [41] model is used to predict the optical flow between the two image crops, which we segment using the segmentation mask and map to correspondences.

### C.2.3 ASpanFormer

AspanFormer [19] is a Transformer-based detector-free model for correspondence estimation. It uses estimated flow maps to adaptively determine the size of the regions within which to perform attention. The latter is done via their proposed Global-Local Attention (GLA) block, which allows them to achieve state-of-the-art performance on a variety of matching benchmarks.

In this project, we use the pre-trained *indoor* model [19] that has been open-sourced by the authors of the paper. The correspondence estimation pipeline begins by cropping segmented RGB images around the segmentation masks. Both cropped images are then resized so that their longer side measures 320 pixels, while preserving the original aspect ratio. For compatibility with the pre-trained model the shorter size is then padded with zeros, resulting in $320 \times 320$ images. Both resized RGB crops are then converted to grayscale images, which are then passed directly as input to the pre-trained model.

### C.2.4 ASpanFormer (FT)

We further fine-tune the pre-trained weights of the ASpanFormer model on an object-centric dataset which we have generated in simulation (see Appendix D.1). We have chosen to do this, as fine-tuning on an object-centric dataset should allow the model to be more in-distribution when dealing with a robot manipulation setting compared to the original ASpanFormer model that was trained on feature-rich scenes with multiple objects.

### C.3 Relative Orientation Estimation Methods

We consider three different methods for predicting the relative object orientation, around the robot's z-axis, from a pair of RGB-D images. These methods include NOPE [18], a recently proposed unseen object relative orientation estimator, and a PointNet++ [52]-based regression and classification models, that we have trained on an object-centric dataset generated in simulation (see Appendix D.1), relying on domain randomisation and data augmentation techniques for sim-to-real transfer [53].

We note that we have tried using NOPE, and training both the regression and classification models to predict full 3DoF relative orientations, but found this to be not very accurate, leading to a poor performance of the final implementation in the real world. Hence, we leverage the prior knowledge that objects are going to be rotated only around the robot's z-axis between the demonstration and deployment to make the problem tractable.

Once a model predicts the relative object orientation, we use a heuristic that applies this rotation to the first partial point cloud and then centres the two partial point clouds to predict the relative translation. We have also experimented with training another PointNet++ [52] for learning a residual correction to this heuristic but did not find this to bring significant improvements.

### C.3.1 NOPE

NOPE is a recently proposed unseen object relative orientation estimator. We use the pre-trained weights provided by the author, trained on 1000 random object instances from each of the following 13 categories from the ShapeNet dataset [54]: airplane, bench, cabinet, car, chair, display, lamp, loudspeaker, rifle, sofa, table, telephone, vessel. During training, NOPE trains a U-Net to predict the embedding of a novel view of an object, given a reference image and a relative pose. Then at inference, it first takes as input a support image of an object and predicts the embedding of that object under many relative orientations, effectively creating templates for template matching. Then given a query image of that same object, NOPE first computes its embedding and then finds the embedding's distance to all the templates, giving a distribution over the possible relative orientations between the query and the support image. The predicted orientation will correspond to the most similar template.

To use NOPE to only predict the relative orientation around the robot's z-axis, we only sample rotation matrices that correspond to relative orientations around the robot's z-axis (which has the same direction as the object's z-axis). To be specific, we create 90 templates corresponding to rotations ranging from $-44.5°$ to $44.5°$ spaced $1°$ apart. NOPE then encodes all of the templates as well as the query object orientation and selects the rotation whose encoding is most similar to that of the query according to the root mean squared error. Once we have the orientation predicted by NOPE, we use the heuristic described at the beginning of this Subsection to estimate the relative translation, completing the process of pose estimation using NOPE.

### C.3.2 Regression

We implement both the regression and the classification baselines to compare simpler approaches trained on an object-centric dataset (see Appendix D.1) to more sophisticated baselines trained on more general datasets, such as DINO, GMFlow and the ASpanFormer. To this end, we implement a Siamese PointNet++ [52] encoder made of three set-abstraction layers and three linear layers, for a total of $\sim$1.8 M parameters. The encoder independently encodes the point clouds of a target object obtained from the demonstration and the test scene. The two output embeddings are then concatenated into a single 512-dimensional vector that we feed as input into a 3-layer perceptron to fuse the information together and regress the object's rotation magnitude around the robot's z-axis. More specifically, the network returns a normalised rotation, where $-1$ and $1$ correspond to $-45°$ and $45°$ respectively. This multilayer perceptron is composed of layers with 256, 128 and 1 hidden node respectively, summing up to $\sim$0.2 M parameters, for a total model size of $\sim$2 M parameters. The model was trained with the ADAM [55] optimiser and the mean squared error loss.

The input point clouds to the Siamese PointNet++ are expressed in the robot's frame, have a zero mean, and are downsampled to 2048 points. The features of each of the points include the point position and colour. Since point positions are used as point coordinates in the PoinNet++ architecture, including them as additional features may seem like including redundant information. However, we have found that doing so improves performance in practice. The most likely reason for this is that PoinNet++ uses point coordinates to cluster points and aggregates features for the different clusters. Without including point positions as features, this information would not be explicitly used to derive the global point cloud feature vector.

### C.3.3 Classification

This baseline is equivalent to the regression method described above, with the exception that the last layer of the 3-layer perceptron does not regress the rotation but instead outputs a probability distribution over 90 possible classes, where each class represents an angle between $-44.5°$ and $44.5°$, equally spaced $1°$ apart. This model has $\sim$2 M parameters and was trained with the ADAM [55] optimiser using the binary cross entropy loss.

## Appendix D    Datasets

We begin this section by describing the object-centric dataset we have generated to train the regression and classification models, and to fine-tune the ASpanFormer model (Appendix D.1). We then describe the dataset we have generated to benchmark all considered unseen object pose estimation methods in simulation (Appendix D.2). Finally, we describe the real-world dataset we used to determine when to stop training the regression and classification models to bridge the sim-to-real gap (Appendix D.3).

### D.1    Training Dataset

The object-centric dataset used to train the regression and classification models, and to fine-tune the ASpanFormer model, was generated using Blender [38] and consists of $\sim$ 226K image pairs. For each image pair, we first create a scene and import a random object from either the ShapeNet dataset [54] or the Google Scan Objects dataset [37] The object is imported at a random pose and there is a $90\%$ probability that its texture will also be randomised, by changing its colour and material properties, such as reflectivity. We then randomise the number, position, energy and strength of external light sources and render an RGB-D image and segmentation mask of the object.

After that, we perturb the object by rotating it around the world's z-axis (perpendicular to the floor) by a maximum of $45°$ either clockwise or counterclockwise and randomly displacing it somewhere

within the visible scene. Finally, we again randomise the position, energy and strength of the external light sources, and render another RGB-D image and segmentation mask. We conclude by recording the relative object pose between the two scenes. Additional data generation details are summarised in Table 3 and some examples of generated image pairs can be seen in Figure 8.

| Characteristic | Randomisation |
| --- | --- |
| Camera Extrinsic | In simulation, we set the camera extrinsic matrix equal to that for our real-world setup. However, to account for possible calibration errors, for each scene in simulation, we randomly perturb the extrinsic matrix by a maximum of 1cm in translation and 2° in orientation. |
| Object Type | Objects are chosen from ShapeNet with a probability of 85% and from Google Scan with 15% probability. Within these two families, the objects are sampled uniformly, but avoiding categories that were excessively out of distribution for object manipulation, such as airplanes, pistols or watercrafts. |
| Object Position | The object position is randomised by an arbitrary magnitude as long as the object remains fully visible to the camera. We use rejection sampling to ensure this. |
| Object Orientation | Once a random object pose is generated to capture the first image, the object's orientation is changed by a random rotation between $-45°$ and $45°$ around the world's z-axis. |
| Object Texture | The appearance of each different component of the object's model gets randomised with a 10% probability. In particular, if randomised, in 20% of cases, the component's colour is set to be monochromatic and the colour could be any of the following, according to these probabilities: normal (35%), dark (15%), very dark (5%), bright (15%), very bright (5%), pale (15%), dark pale (5%), bright pale (5%). If not monochromatic, with 20% probability the randomisation is applied as a texture, where the latter is sampled uniformly from either the Haven [56] or MIL [11] texture datasets. Additionally, the material properties get randomised as well. More specifically with 50% probability the roughness of the material gets changed, with 30% probability the metallic properties get altered, with 30% probability the specularity properties, with 30% probability the material's anisotropy, with 15% probability its sheen, and finally with 5% probability the clearcoat property of the material is varied. |
| Lighting | When generating a scene, the lighting conditions get randomised. There are three main modes the light can be in, whose parameters get further randomised. More specifically, the five modalities with their probabilities are: mostly ambient (5%), strong top shadow (30%), generic shadow (30%), very bright (5%), very dim (30%). Once the modality has been sampled, aside from the "strong top shadow" mode, either one or two light sources get placed in the scene. The light location, energy and ambient strength get then randomised for each of the light sources independently, creating very diverse lighting conditions. |

Table 3: Detailed explanation of the various randomisation strategies applied for the process of data generation.

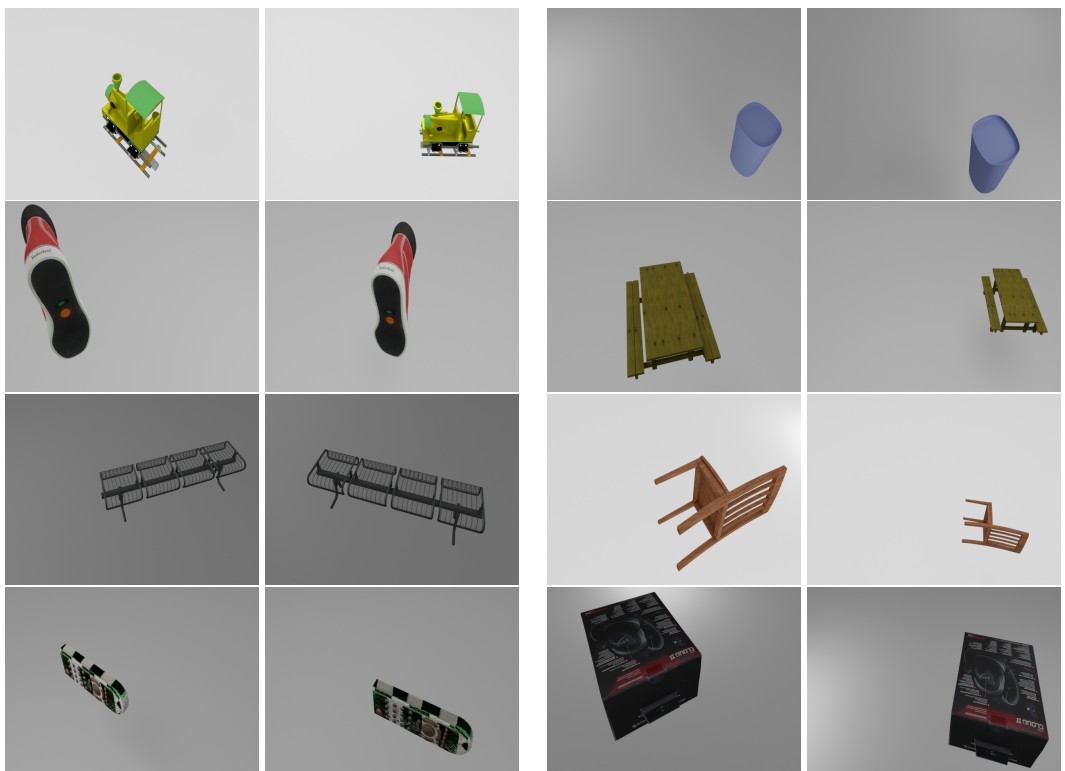

Figure 8: Eight examples of image pairs generated in simulation. Each row contains two image pairs, showing the same object with a random texture and colour. The object position and orientation, as well as lighting conditions are randomised between the two images. The background is not randomised as the objects will later be segmented out, making the choice of background irrelevant.

### D.2   Evaluation Dataset

In Section 4.2 of the main paper, we evaluate the performance of eight different one-shot unseen object pose estimation methods in simulation. To evaluate these methods, we generated a separate dataset to that used for training, using objects that the ASpanFormer, regression and classification models were not trained on. We divided these objects into five categories, briefly explained hereafter. Examples for each of the categories can be found in Figure 9

1. **Non-Symmetrical** (Non-Sym.) are objects that are not symmetric around any of their axes. This category has 25 objects.
2. $\infty$**-Symmetrical** ($\infty$-Sym.) are objects that have an infinite degree of symmetry around their z-axis. This category has 10 objects.
3. $\infty$**-Symmetrical Geometry** ($\infty$-Sym. Geo.) are objects whose geometry has an infinite degree of symmetry around the object's z-axis, but have a non-symmetric texture.
4. $N$**-Symmetrical** ($N$-Sym.) are objects which have a finite degree of symmetry around their z-axis. This category has 10 objects. For instance, a cube has a rotation symmetry of order 4 around its z-axis.
5. $N$**-Symmetrical Geometry** ($N$-Sym. Geo.) are objects which have a non-symmetrical texture but whose geometry has a finite degree of symmetry around the object's z-axis. This category has 5 objects.

Examples of the objects that could be found in each category are the following. 1) **Non-Symmetrical**: kettles, mugs, shoes, caps, etc. 2) $\infty$**-Symmetrical**: ramekins, bowls, vases, cups, etc. 3) $\infty$**-Symmetrical Geometry**: cans, tape, cylindrical medicine packages, etc. 4) $N$**-Symmetrical**: square plates, square bowls, boxes, chests, etc. 5) Lastly $N$**-Symmetrical Geometry**: non-uniform boxes, sponges, cylindrical speakers, etc.

Overall, for each different object, we generated 20 image pairs, resulting in a total dataset size of $1,100$ image pairs.

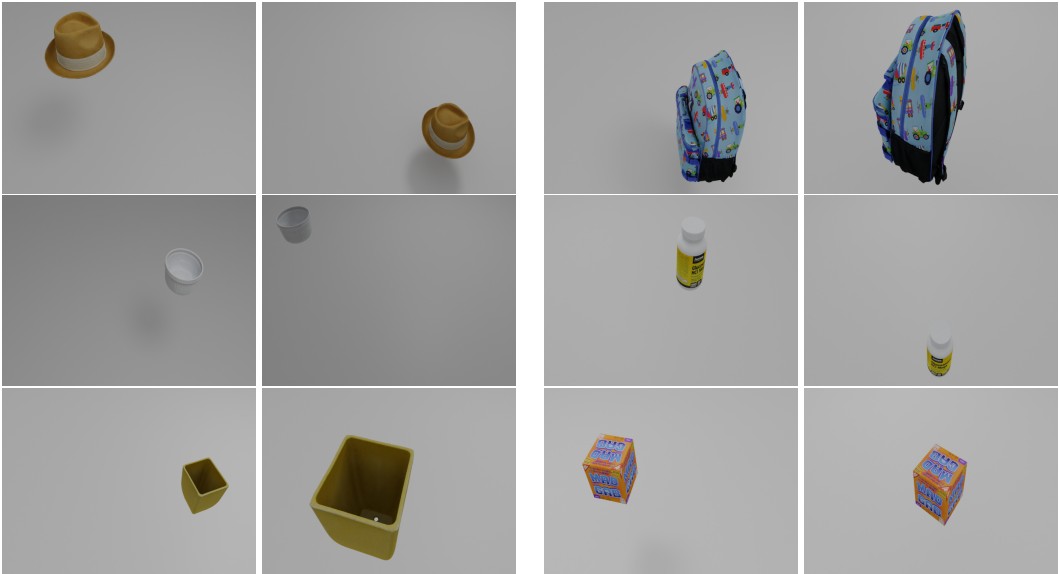

Figure 9: Examples of image pairs generated for the evaluation dataset. The top row shows two image pairs for non-symmetric objects. Second row shows image pairs for (left) an $\infty$-Symmetrical object and (right) an $\infty$-Symmetrical Geometry object. Bottom row shows image pairs for (left) an $N$-Symmetrical object and (right) an $N$-Symmetrical Geometry object.

### D.3 Real-World Validation Dataset

We collect a real-world validation dataset to obtain a criteria for early stopping when training the regression and classification models. To this end, we collect 71 image pairs of 7 different everyday objects, including two different toasters, one pan, one pot, a wooden box, a small black box and a rigid plastic container. We collect the images mimicking the data generation process in simulation. We firstly place an object on the table along with a removable AprilTag [57], which allows us to initially record the ground truth pose and then remove the tag to collect the desired RGB-D image without the tag being visible. Subsequently, we perturb the object's pose following the same strategy as with the simulated scenes, and we record another pose and RGB-D image. Examples of the collected data can be found in Figure 10. When training a model, we monitor its pose estimation error on this dataset to determine when to stop the training.

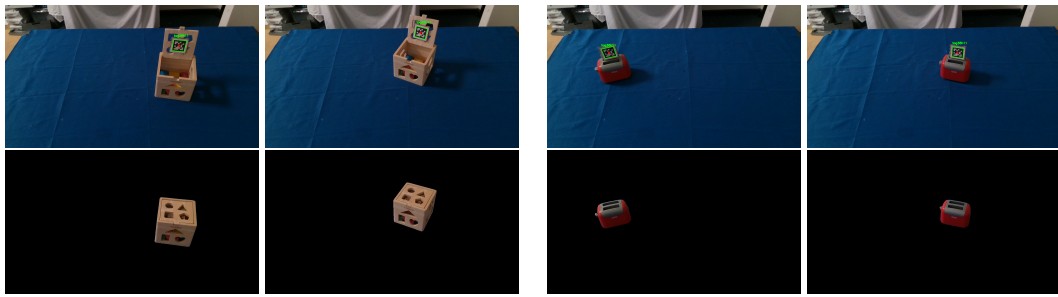

Figure 10: The left and right double columns show pairs of images for a wooden box and a red toaster respectively. Within one image pair, to go from one picture to the other the object has been randomly translated and then randomly rotated. The first row illustrates how the ground truth relative pose has been determined, that is through the use of AprilTags. These were carefully placed so that they could be hidden without affecting the pose of the object, allowing for the capture of the images shown in the second row.

| | Non-Sym. | $\infty$-Sym. | $\infty$-Sym. Geo. | $N$-Sym. | $N$-Sym. Geo. | Mean |
|---|---|---|---|---|---|---|
| Class. | $8.4 \pm 14.5$ | $\mathbf{1.6 \pm 0.8}$ | $1.9 \pm 1.8$ | $\mathbf{6.7 \pm 12.3}$ | $3.9 \pm 4.8$ | $5.9 \pm 11.2$ |
| ASpan. (FT) | $\mathbf{8.1 \pm 16.3}$ | $3.9 \pm 6.4$ | $3.2 \pm 3.9$ | $6.9 \pm 13.8$ | $\mathbf{2.0 \pm 4.7}$ | $6.0 \pm 13.0$ |
| Reg. | $13.7 \pm 17.8$ | $1.5 \pm 2.8$ | $\mathbf{1.2 \pm 0.8}$ | $12.2 \pm 15.3$ | $9.7 \pm 14.7$ | $9.8 \pm 15.3$ |
| DINO | $15.1 \pm 21.5$ | $6.8 \pm 7.9$ | $3.7 \pm 3.7$ | $13.3 \pm 20.2$ | $5.9 \pm 13.3$ | $11.3 \pm 18.2$ |
| Aspan. | $15.3 \pm 21.5$ | $7.2 \pm 8.7$ | $5.5 \pm 6.4$ | $12.6 \pm 16.9$ | $6.5 \pm 11.0$ | $11.5 \pm 17.4$ |
| NOPE | $25.3 \pm 16.9$ | $1.8 \pm 3.9$ | $1.9 \pm 1.7$ | $24.5 \pm 15.4$ | $23.1 \pm 15.8$ | $18.8 \pm 17.4$ |
| ICP | $22.6 \pm 40.1$ | $3.7 \pm 6.2$ | $3.8 \pm 14.2$ | $9.4 \pm 17.3$ | $14.3 \pm 29.3$ | $14.3 \pm 30.8$ |
| GMFlow | $30.9 \pm 23.7$ | $21.7 \pm 26.1$ | $20.9 \pm 15.5$ | $30 \pm 26.2$ | $31.5 \pm 25.2$ | $28.4 \pm 24.6$ |
| Mean | $16.6$ | $4.4$ | $3.9$ | $13.5$ | $10.3$ | |

Table 4: Full results for the simulation benchmarking experiments regarding translation errors in pose estimation, expressed in centimetres.

| | Non-Sym. | $\infty$-Sym. | $\infty$-Sym. Geo. | $N$-Sym. | $N$-Sym. Geo. | Mean |
|---|---|---|---|---|---|---|
| Class. | $\mathbf{6.7 \pm 13.1}$ | $\mathbf{0.2 \pm 0.1}$ | $0.4 \pm 1.4$ | $\mathbf{5.4 \pm 10.7}$ | $\mathbf{2.1 \pm 6.3}$ | $4.3 \pm 10.1$ |
| ASpan. (FT) | $8.6 \pm 17.1$ | $3.3 \pm 5.8$ | $2.6 \pm 3.1$ | $6.7 \pm 13.3$ | $2.2 \pm 6.2$ | $6.1 \pm 13.4$ |
| Reg. | $13.7 \pm 18.0$ | $0.7 \pm 2.9$ | $\mathbf{0.3 \pm 0.1}$ | $11.4 \pm 13.9$ | $9.3 \pm 14.1$ | $9.4 \pm 15.3$ |
| DINO | $16.4 \pm 22.9$ | $5.7 \pm 7.0$ | $3.1 \pm 3.0$ | $13.2 \pm 19.6$ | $5.8 \pm 12.4$ | $11.6 \pm 18.9$ |
| ASpan | $16.8 \pm 24.2$ | $5.7 \pm 6.7$ | $4.6 \pm 5.4$ | $11.5 \pm 15.5$ | $6.5 \pm 11.1$ | $11.7 \pm 18.8$ |
| NOPE | $25.4 \pm 16.1$ | $0.9 \pm 3.9$ | $0.4 \pm 1.4$ | $23.8 \pm 14.4$ | $23.2 \pm 15.7$ | $18.4 \pm 17.2$ |
| ICP | $24.1 \pm 43.4$ | $2.6 \pm 3.9$ | $3.2 \pm 11.8$ | $8.4 \pm 16.0$ | $13.1 \pm 25.9$ | $14.4 \pm 32.4$ |
| GMFlow | $33.3 \pm 27.7$ | $17.0 \pm 18.9$ | $17.9 \pm 13.2$ | $28.1 \pm 22.3$ | $27.8 \pm 21$ | $27.6 \pm 24.5$ |
| Mean | $16.4$ | $4.2$ | $3.7$ | $13.3$ | $10.05$ | |

Table 5: Full results for simulation benchmarking experiments regarding rotation errors in pose estimation, expressed in degrees.

## Appendix E    Benchmarking One-Shot Unseen Object Pose Estimators

In Section 4.2 of our main paper, we benchmark the eight unseen object pose estimation baselines introduced in Appendix C, on the simulated dataset described in Appendix D.2.

**Error Definition** Consider a ground truth relative object pose between a pair of images, $^{R}\boldsymbol{T}_{\delta}$, and a pose estimate $^{R}\bar{\boldsymbol{T}}_{\delta}$. We begin by calculating the transformation $\boldsymbol{T}_{\epsilon} = [\boldsymbol{R}_{\epsilon} | \boldsymbol{t}_{\epsilon}]$ that maps the pose estimate to the ground truth pose, i.e.

$$^{R}\boldsymbol{T}_{\delta} = \boldsymbol{T}_{\epsilon} \, ^{R}\bar{\boldsymbol{T}}_{\delta}$$
$$\rightarrow \boldsymbol{T}_{\epsilon} = \, ^{R}\boldsymbol{T}_{\delta} \left( ^{R}\bar{\boldsymbol{T}}_{\delta} \right)^{-1} .$$

We then define the translation and rotation error as

$$t_{\text{error}} = ||\boldsymbol{t}_{\epsilon}||_{2} \qquad\qquad R_{\text{error}} = ||\log\left(\boldsymbol{R}_{\epsilon}\right)||_{2}$$

where log is the logarithmic map for the $SO(3)$ group.

In practice, for object categories with an order of symmetry $> 1$ (see Appendix D.2), there is always more than a single ground truth pose $\boldsymbol{T}_{\delta}$ for each image pair. In those cases, we independently compute the translation and rotation error between the pose estimate and each of the valid relative object poses, and consider the pair of errors corresponding to the ground truth pose that gives rise to the smallest rotation error.

For objects with an infinite degree of symmetry around the z-axis, we create 360 ground truth relative object poses for each image pair. That is, we discretise the possible rotations around the z-axis into 360 bins, and find a relative object pose corresponding to each possible relative rotation. For objects with a degree of symmetry of 4 around the z-axis, we create 4 possible relative object poses per image pair. Finally, for objects with a degree of symmetry of 2 around the z-axis, we create 2 possible relative object poses per image pair.

**Results** In Table 4 we show the full results concerning the translation error in pose estimation for the individual object categories discussed in Appendix D.2. Similarly, we do the same for rotation errors in Table 5.

**Discussion** From Tables 4 and 5, we can clearly see that on average, the higher the degree of symmetry of an object, the lower the error in the predictions of all methods. This is intuitive, as the larger the order of symmetry of an object, the larger the possible set of correct relative pose labels.

Surprisingly, predicting the rotation for the symmetrical geometry categories has turned out to be easier than for the corresponding categories where the visual textures are symmetric as well. However, as we have only considered 5 and 10 objects each for these categories, these results may not be statistically significant.

## Appendix F   Trajectory Transfer Implementation Details

### F.1   Incorporating an Inductive Bias

As mentioned in Appendix C.3, when training the regression and classification models, and when using NOPE [18] to predict 3DoF relative orientations, we found high pose estimation errors that compromised real-world performance. Hence, we have chosen to train the regression and classification networks, and to use the pre-trained NOPE model, only to predict the relative object orientation around the robot's z-axis. This design choice is motivated by the fact that for most manipulation tasks, a test object translates in 3DoF while being rotated only around the world's z-axis between the demonstration and deployment, and the world's and robot's z-axes are aligned.

To ensure a fair comparison between regression, classification, NOPE, and the remaining considered baselines, we also incorporate this predictive bias into their predictions. Specifically, consider a relative pose estimate expressed in the robot frame (see Appendix A):

$$^{R}\boldsymbol{T}_{\delta} = \left[^{R}\boldsymbol{R}_{\delta} | ^{R}\boldsymbol{t}_{\delta}\right],$$

where $^{R}\boldsymbol{R}_{\delta} \in SO(3)$ is the relative orientation prediction and $^{R}\boldsymbol{t}_{\delta} \in \mathbb{R}^3$ is the relative translation prediction. To incorporate the inductive bias that the object only rotates around the robot's z-axis into such an estimate, we first convert $^{R}\boldsymbol{R}_{\delta}$ to Euler angles, set rotations around the non-z axes to zero, and convert back to a rotation matrix $^{R}\tilde{\boldsymbol{R}}_{\delta}$. Now, given $^{R}\tilde{\boldsymbol{R}}_{\delta}$ and an end-effector pose $\boldsymbol{T}_{RE}^{Demo}$ that we would like to align with an object at test time via trajectory transfer (see Equation 5 of the main paper), we adjust the translation component of $^{R}\boldsymbol{T}_{\delta}$ to account for the modification of $^{R}\boldsymbol{R}_{\delta}$ using the equation:

$$^{R}\tilde{\boldsymbol{t}}_{\delta} = {}^{R}\boldsymbol{R}_{\delta}\boldsymbol{t}_{RE}^{Demo} - {}^{R}\tilde{\boldsymbol{R}}_{\delta}\boldsymbol{t}_{RE}^{Demo} + {}^{R}\boldsymbol{t}_{\delta},$$

where $^{R}\tilde{\boldsymbol{t}}_{\delta}$ is the adjusted relative translation, and $\boldsymbol{t}_{RE}^{Demo}$ is the translation component of $\boldsymbol{T}_{RE}^{Demo}$. The intuition behind this equation is that it ensures that the position of the end-effector after trajectory transfer with a modified pose estimate $^{R}\tilde{\boldsymbol{T}}_{\delta} = [^{R}\tilde{\boldsymbol{R}}_{\delta}|^{R}\tilde{\boldsymbol{t}}_{\delta}]$ is the same as the position that would have been obtained when using the non-modified pose estimate $^{R}\boldsymbol{T}_{\delta}$.

### F.2   Aligning the Full Trajectory

In theory, we could use trajectory transfer (see Equation 5 of the main paper) to solve for the full end-effector trajectory aligned with the object at the deployment pose and could track this trajectory using trajectory tracking. However, in practice, we have found that this resulted in very jerky trajectories. Hence, instead, we use trajectory transfer to only align the first end-effector pose of the demonstration with the deployment scene, and send the robot to that pose using inverse kinematics. From there, we align the full end-effector trajectory with the demonstrated trajectory by replaying the demonstrated end-effector velocities expressed in the end-effector frame. Although the robot does not realise these velocities instantaneously, in practice, we have found this to work sufficiently well and to perform better than using a trajectory tracking system.

## Appendix G   Sensitivity to Non-Geometric Noise

In this section, we focus on trajectory transfer using regression for unseen object pose estimation, which was the best-performing method in our real-world experiments (see Section 4.3 of our main paper), and analyse its sensitivity to distractors and changes in lighting conditions and backgrounds.

### G.1   Sensitivity to Changes in Lighting Conditions

To investigate the robustness of trajectory transfer to changes in lighting conditions, we follow the same experimental procedure as described in Section 4.3 of our main paper. That is, we divide a $30 \times 75$cm region on a table in front of the robot into ten quadrants measuring $15 \times 15$cm and use the demonstrations collected when carrying out the main experiment to facilitate a direct comparison to the remaining results presented in the main paper.

At test time, for each quadrant, we randomly perturb the position, luminosity and colour temperature of an external LED light source (see Figure 11 for examples), and randomly place the test object

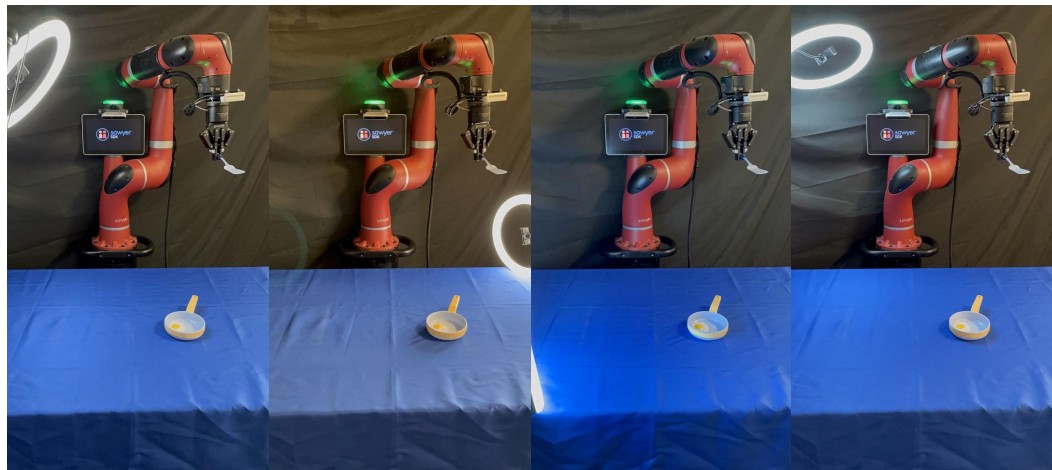

Figure 11: Examples of how the position, luminosity and colour temperature of an external LED light source have been randomly perturbed between different evaluations to study the robustness of trajectory transfer using regression for unseen object pose estimation to changes in lighting conditions.

within that quadrant with a random orientation between $\pm 45°$ of the demonstrated orientation. This results in ten evaluations per task.

|  | Plug | Pot | Toaster | Dishwasher | Mug | Egg | Bottle | Tea | Bowls | Can | Mean |
|---|---|---|---|---|---|---|---|---|---|---|---|
| Fixed Lighting | **20** | **30** | **90** | **70** | **100** | **90** | **100** | **100** | 80 | **100** | 78 |
| Changes in Lighting | 10 | 20 | 60 | **70** | **100** | 70 | 80 | **100** | 90 | **100** | 70 |

Table 6: Real-world success rates (%) of TT (Reg.) averaged over ten trials under fixed lighting conditions and changes in lighting conditions.

The results from this experiment are shown in Table 6. For reference, this table also includes the results for trajectory transfer using regression under fixed lighting conditions from our main experiment which is described in Section 4.3 of our main paper. As these results illustrate, trajectory transfer using regression displays a strong performance as lighting conditions are randomised between the demonstration and test scene, with an average decrease in performance of only $8\%$. We attribute the strong performance of this baseline under changes in lighting conditions to the fact that the dataset used to train this baseline randomises lighting conditions between the two input images, alongside the hue, value and saturation of the two images, as part of domain randomisation (see Appendix C.3 and D.1).

## G.2 Sensitivity to Distractors and Changes in Background

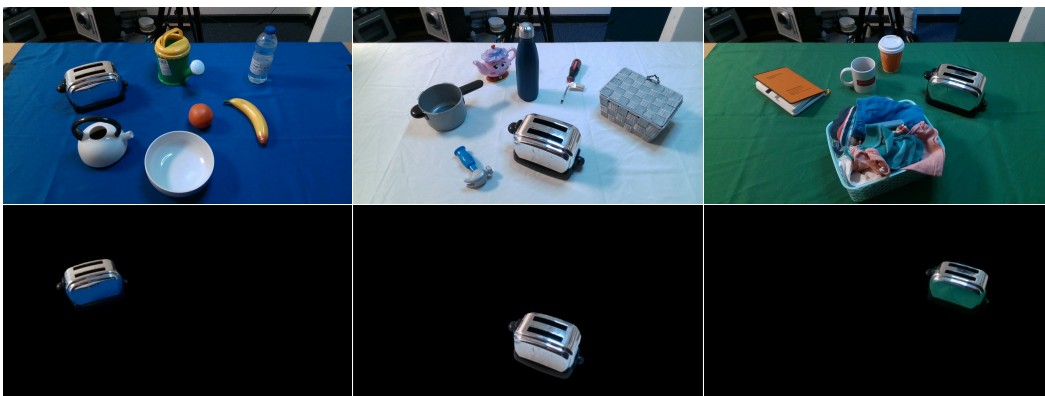

Figure 12: Example of the combined performance of Owl-ViT and SAM when segmenting a toaster in cluttered scenes with different backgrounds. By isolating only the object of interest, the chosen unseen object pose estimator is unaffected by the mentioned changes.

Our formulation of trajectory transfer using unseen object pose estimation assumes segmented input RGB-D images. Hence, the robustness of the framework to distractors and changes in the background is only dependent on the used segmentation pipeline and not on the backbone pose estimator itself. In our implementation, we use a combination of OWL-ViT [45] and SAM [46] for the segmentation pipeline. That is, we first query OWL-ViT for a bounding box of the test object using a language prompt *"An image of a X"*, where $X$ is the category of the considered test object (e.g. can, mug, toaster, plug etc.). We then crop the RGB-D image using the output bounding box and pass the cropped RGB image to SAM to obtain a segmentation mask of the target object. With this pipeline, we have observed consistent good performance even under clutter. Examples of segmentations of the toaster in three different cluttered scenes with three different backgrounds can be seen in Figure 12.

