# OpenReview forum: "One-Shot Imitation Learning: A Pose Estimation Perspective"
_robot-learning.org/CoRL/2023/Conference — CoRL 2023 Poster_

### Official Review · Reviewer_DHE8 · 2023-07-10

**Confidence:** 3
**Originality:** Good
**Technical Quality:** Good
**Clarity Of Presentation:** Fair
**Impact:** 3

**Recommendation:**

Weak Accept: I recommend accepting the paper, but will not argue for my recommendation if the majority of other reviewers have a different opinion.

**Review:**

Strenghts:
- The paper presents a simple approach that is able to render impressive results in real-world experiments. I think the paper can be useful for the community.
- The paper presents extensive comparisons in simulation with 8 different backbones for pose estimation, which are carefully chosen, implemented and tuned.

Weaknesses:
- There seems to be little technical novelty in this work.
-  The clarity of the paper can be significantly improved. The only part where an overview the method is clearly explained is in the caption of Fig. 2. In my opinion, the abstract and introduction don't address the contributions or the main methodology of the paper, but rather focus on why the problem is hard. The paper presents more prominently the four conducted investigations than the method itself, and presents the method as a byproduct of that. In my opinion, this structure could be clearer.
- Even if the method presented seems to be quite simple, it shows that is effective. Is this the first method that poses the one shot imitation learning problem as a trajectory alignment problem? If so, mention it in the introduction.
- It's not clear if the robot gets only one picture at the beginning of the deployment phase (open-loop) or if it constantly gets pictures until the end effector is aligned with the demonstration picture (closed-loop). Would be beneficial to explain this in the manuscript.
- Section 4 lacks a clear motivation. These ablation studies seem to be one important contribution of the paper, but however are not motivated well enough in the section.
- In Section 4, the order in which Fig. 3 and 4 appear compared to the order in which these figures are referenced seems to be swapped. Fig. 4 seems to be connected with the first part of Section 4, and this can be confusing for the reader.
- Reading Table 1 and Table 2 can give the impression that this paper is a benchmark paper. Would maybe be easier to understand the contributions if the paper is presented as so.
- I wonder why in Section 5.2 an ablation with respect to the rotation of the object (and not only the translation) is not presented.



**Quality Of The Limitations Section:**

Limitations are addressed clearly

**Questions For Rebuttal:**

The paper can be improved significantly, specially in terms of clarity and structurally (as written in the weakness part of my review). I would rather write this paper as a benchmark paper than as a contributing paper, since the technical contribution is very limited.

**Robotics Focus:**

Sufficient demonstration on hardware

**Summary Of Paper:**

In this paper the authors aim to solve the problem of relative pose estimation given RGB-D images. The approach is imitation-based and it's constrained to learn only from one single demonstration per task.  A demonstration is defined as an RGB-D image and an end-effector trajectory. The problem is modeled as trajectory alignment problem and boiled down to finding a transformation matrix between the demonstration and the deployment. Intuitively, once the pose is estimated, the robot can "repeat" the demonstration once the estimated relative pose is applied. The paper is accompanied by impressive real world videos.

**Summary Of Recommendation:**

The paper presents impressive real-world results for the task of teaching a manipulation robot how to do a task with only one demonstration per task. The problem of learning a task is posed as a trajectory alignment problem, and many baselines are compared for solving it. I believe this paper can be useful for the community once the authors address the structural flaws I highlight in my review.

---

### Official Review · Reviewer_NRUS · 2023-07-17

**Confidence:** 3
**Originality:** Fair
**Technical Quality:** Fair
**Clarity Of Presentation:** Very Good
**Impact:** 3

**Recommendation:**

Weak Accept: I recommend accepting the paper, but will not argue for my recommendation if the majority of other reviewers have a different opinion.

**Review:**

*Quality*

Pros
- The paper performed a comprehensive analysis of 8 relative pose estimation methods. For each method, the paper reports both the pose estimation error and the success rate when deployed on a real robot.
- The experiments covers 10 different tasks that covers diverse object interactions. 10 evaluation trials are performed for each task times each pose estimation method. However, the tasks all share similar attributes and may not reflect real-world challenges. See next paragraph for more detail.

Cons
- The proposed framework is incremental to previous works e.g. DOME cited in the paper. The authors did a fair comparison on page 2 line 67, and acknowledges that:

    "The main difference [...] is that they focus on top-down settings and rely on a wrist-mounted camera. On the contrary, we [...] rely on an external camera, allowing the robot to constantly have a complete view of the workspace that cannot get occluded by held objects."

    **Reducing the need of a RGB wrist camera while requiring a third-person RGB-D camera is not sufficient to motivate the new method.** The author also does not provide empirical comparison, or any detailed discussion, to justify why the proposed relative pose estimation approach is preferred compared to the visual-servoing approach of previous work.
- **The proposed framework is limited to trajectories that involve a single (moving) object**, which hinders its usefulness in real-world scenarios. If there are two objects of interest in a single trajectory, it is unclear how the two transformations can be used to align the trajectory. The paper also does not include a explicit discussion of this limitation.

*Clarity*

The presentation is clear.

*Originality*

The work is original, though limited novelty.

*Significance*

The empirical analysis is substantial in evaluating ablations of the proposed method, while it leaves out comparisons to previous works that tackle similar problems.

**Quality Of The Limitations Section:**

Additional details required

**Questions For Rebuttal:**

*For the first point under Quality -> Cons:*
Provide a more thorough empirical comparison or discussion compared to previous methods such as DOME. As noted in the previous section, the proposed method requires better motivation and justification.

*For the second point under Quality -> Cons:*
Provide a more detailed discussion of this limitation.

**Robotics Focus:**

Sufficient demonstration on hardware

**Summary Of Paper:**

The paper proposes a new framework for one-shot imitation learning. Specifically, the demonstration is composed of one third-person image of the scene at the beginning of the episode, and also the end-effector trajectory. At test time, the object may be placed at a different position and the robot should still perform the task.

The approach authors proposed is to perform relative object pose estimation between the test time image and the demo image, i.e. predict a homogeneous transformation from two RGB-D images. The transformation is then used to move the robot to a similar relative starting pose w.r.t the object of interest, after which replaying the demo should accomplish the task.

**Summary Of Recommendation:**

I recommend weak reject as this work lacks comparisons to related works and is not well motivated, despite relatively thorough ablations.

Updated to weak accept after rebuttal.

---

### Official Review · Reviewer_Sjn1 · 2023-07-23

**Confidence:** 5
**Originality:** Poor
**Technical Quality:** Fair
**Clarity Of Presentation:** Very Good
**Impact:** 1

**Recommendation:**

Weak Accept: I recommend accepting the paper, but will not argue for my recommendation if the majority of other reviewers have a different opinion.

**Review:**

# Strengths
## Task-centric evaluation of one-shot pose estimation methods
This paper evaluates several one-shot pose estimation methods over 10 different tasks. Many of these methods are only evaluated on pose or pixel level metrics. These metrics rarely directly map to end performance for robotic applications.

# Weaknesses
## Limited analysis
The analysis done on the effects of different types of errors on trajectory alignment tasks are limited. The effects of the noise analyzed in this paper are transforming errors though a chain of transformations. This does not give us a better understanding of what types of errors actually trigger failures in each type of problem. The spatial generalization analysis in section 5.2 is more along the lines of the analysis that would be useful, but it is relatively light in scope. Additionally, robustness to non-geometric noise, such as lighting was not analyzed.

(Post Rebuttal) Authors greatly improved the analysis of the effects of pose errors and lighting changes on a variety of tasks.

## No novelty
The described “new framework” is a PointNet++ used to regress or classify relative pose. This is basically a simplified version of DCP [1]. Without further details I can’t be sure how novel this approach is. I do understand that this is meant to be an analysis paper, novelty is less crucial, but it should be noted.

[1] Wang, Yue, and Justin M. Solomon. "Deep closest point: Learning representations for point cloud registration." Proceedings of the IEEE/CVF international conference on computer vision. 2019.

**Quality Of The Limitations Section:**

Limitations are addressed clearly

**Questions For Rebuttal:**

See review.

**Robotics Focus:**

Sufficient demonstration on hardware

**Summary Of Paper:**

This paper studies the one-shot imitation learning problem as described as a zero-shot pose estimation and trajectory alignment problem. The authors study the effects of camera extrinsics calibration as well as pose estimation error on end effector error and evaluate several methods of zero-shot pose estimation over ten manipulation tasks. In this study, they train a regression and classification neural network to estimate the relative pose between two objects. In the study, these classification and regression models performed best over most tasks. Additionally, a spatial error analysis was performed over all methods to determine the correlation between object location on the table and task success. This showed that task success was correlated with proximity to the demonstration location, in the center of the table.

**Summary Of Recommendation:**

While I do think that evaluating one-shot pose methods on actual tasks gives a better understanding of their utility, I think the analysis beyond which worked best was lacking. A better analysis of what types of pose errors caused tasks failures and a more thorough look into each tasks error tolerance would have given the reader a better understanding of how to correlate estimator error and task success. With the limited scope of analysis and no added novelty, I can not recommend acceptance.

(Post Rebuttal) With the new analysis, this work could be used to evaluate the accuracies needed for future pose estimation to complete these tasks. I raise my recommendation to weak accept.

---

### Official Review · Reviewer_cUKd · 2023-07-23

**Confidence:** 4
**Originality:** Good
**Technical Quality:** Good
**Clarity Of Presentation:** Good
**Impact:** 3

**Recommendation:**

Weak Accept: I recommend accepting the paper, but will not argue for my recommendation if the majority of other reviewers have a different opinion.

**Review:**

Strengths:
The paper addresses a very important problem in robot learning from demonstration. The generalization of learnt policy in new situations is formulated as A Pose Estimation Perspective with interesting insights on sensitivity analysis of related factors.

The experiments on real robot demonstrate the validity of the approach

Weakness:
The significance of the proposed is not so well placed with respect to the state-of-the-art. The experimental test cases are relatively simple (linear top down motion)


**Quality Of The Limitations Section:**

Limitations are addressed clearly

**Questions For Rebuttal:**

The paper should compare and establish some quantitative metric for comparison against established pose based skill learning in robotics e.g.
 B. Wen, W. Lian, K. Bekris, and S. Schaal, “You Only Demonstrate Once: Category-Level Manipulation from Single Visual Demonstration,” in Proceedings of Robotics: Science and Systems, New York City, NY, USA, June 2022.
E. Valassakis, G. Papagiannis, N. Di Palo, and E. Johns, “Demonstrate once, imitate immediately (dome): Learning visual servoing for one-shot imitation learning,” in 2022 IEEE/RSJ International Conference on Intelligent Robots and Systems (IROS). IEEE, 2022, pp. 8614–8621.
And additionally also distinguish it from the other skill generalization like DMPs and TP-GMM which also uses pose based generalization of policies.


**Robotics Focus:**

Sufficient demonstration on hardware

**Summary Of Paper:**

The paper phrases one-shot imitation learning as an unseen object pose estimation problem which allows use of a single demonstration to learn a new task. With only a single demonstration and no prior knowledge about the object the robot is interacting with, the optimal imitation can be considered to be where the robot and object are aligned in the same way as during the demonstration, throughout the task. Although the observation is appreciable, I guess this is already encoded in many policy generalization schemes via modulation of demonstrated policy based on observations. The interesting sensitivity analysis affecting the performance of the generalization is worth noting.

**Summary Of Recommendation:**

The paper has interesting insights about the pose perspective generalization of policy.
However, I would like to see some amendments as mentioned in my comments above.

---

### Author Response · Authors · 2023-08-14
**General Comment**

Thank you very much to all reviewers for your time in reading the paper, for asking some important questions, and for suggesting some new experiments. We are pleased to hear that our work had “interesting insights about the pose perspective generalization of policy” (cUKd), that “evaluating one-shot pose methods on actual tasks gives a better understanding of their utility” (Sjn1), that we “performed a comprehensive analysis” (NRUS) and that “the paper can be useful for the community” (DHE8).

We have responded to each reviewer individually, where we also upload the updated paper, and below are the main updates and new experiments:

- **We have improved the clarity of the paper’s main objective.** There was a shared feeling that the paper’s objective could be made more clear. As a result, we have edited the paper to highlight the true scope of our work. We now carefully present the paper as an in-depth study on the utility of unseen object pose estimation in the context of one-shot imitation learning. We emphasised that we are not proposing a novel framework as such, but instead perform an investigation on the influence of different factors in the application of unseen object pose estimation to imitation learning. All of our baselines are there to give an intuition on the performance that can be expected from state-of-the-art unseen object pose estimators. However, please note that the combination of one-shot imitation learning and unseen object pose estimation is still novel, and as we show, leads to highly efficient imitation learning.
- **New experiments: we now benchmark against DOME [1].** After the suggestions of some reviewers, our paper now better discusses how our formulation stands when compared to the state-of-the-art in one-shot imitation learning. As such, we have improved the related work section, as well as empirically benchmarked against DOME on all our real-world tasks, after receiving the code and network weights from the authors.
- **New experiments: we now contextualise the noise sensitivity experiments with respect to actual task performance.** We now discuss how errors in calibration and pose estimation directly correlate to success rates in real-world tasks. To do so, we performed additional experiments on task tolerance, and combined the results with the previous sensitivity analysis. This figure ([anonymous link](https://tinyurl.com/mut6tvaa)) summarises our findings, and is discussed in Sections 4.1 and 4.2 of the updated paper.

If reviewers have any further questions, then please do ask us before the deadline, and we would be happy to answer them.

[1] E. Valassakis, G. Papagiannis, N. Di Palo, and E. Johns. Demonstrate once, imitate immediately (dome): Learning visual servoing for one-shot imitation learning. In IEEE/RSJ International Conference on Intelligent Robots and Systems (IROS), 2022.

---

> ### Author Response · Authors · 2023-08-15
> **Robustness to Non-Geometric Noise**
>
> Following the suggestion from Reviewer Sjn1, we have also now added new experiments which study the robustness of our one-shot imitation learning formulation, to illumination changes, distractor objects, and changes in the colour of the tablecloth. These are written up in Section 8 of the updated supplementary material (which is attached [here](https://drive.google.com/file/d/1f7DbTF7LtPcOGiW9VV8jrj2tQh9v8WP-/view?usp=sharing)).
>
> So following this, our paper now provides an in-depth study which evaluates the effects of all of the following aspects: **object pose estimation error, camera calibration error, task tolerance, spatial generalisation, illumination changes, distractor objects, and changes in background colour**. Personally we have learned so much from these experiments, and we really think that the community will find these experiments very useful in understanding where the main sources of error still remain for one-shot imitation learning.

---

> > ### Comment · Reviewer_Sjn1 · 2023-08-15
> > **Quantitative results for "8.2 Sensitivity to Distractors and Changes in Background"**
> >
> > Are there quantitative results for the distractors and changes in background experiments or is it a qualitative statement that the use of object segmentation masks makes the method robust to these changes? This assumption may not hold when the distractors partially occlude / pollute the segmentation masks.

---

> > > ### Author Response · Authors · 2023-08-15
> > > **Quantitative results for "8.2 Sensitivity to Distractors and Changes in Background": Answer from Authors**
> > >
> > > Yes, these are qualitative results to illustrate the robustness of state-of-the-art detection and segmentation methods to backgrounds, and thus showing that today's methods are suitable for inclusion in an imitation learning framework based on unseen object pose estimation. It's true that distractors could make detection / segmentation more challenging, but our framework is agnostic to the specific detection / segmentation methods, and we see detection / segmentation in the presence of distractors a pure computer vision challenge which is currently being addressed by the computer vision community. Instead, we focus our experiments on robotics and imitation learning questions, and draw our conclusions about what can be done with imitation learning given today's state-of-the-art in computer vision.

---

### Author Response · Authors · 2023-08-15
**Final 8 page paper and supplementary material**

Dear AC and Reviewers,

Following all the new experiments and re-writing we have done in recent days, we would just like to provide you with a link to our final 8-page paper:  [link](https://drive.google.com/file/d/1_xz38McirYH5OUbgeD2VQ9eB7klgtLxT/view?usp=sharing). The supplementary material is attached as an appendix.

 Thank you,

 The Authors.

---

### Decision · Program_Chairs · 2023-08-30

**Decision:**

Accept (Poster)

**Comment:**

The paper introduces a new method to tackle the challenging problem of one-shot imitation learning, which is formulated as a trajectory alignment problem. After the authors' rebuttal, most concerns of the reviewers have been addressed, and majority reviewers agree on accepting the paper.

Due to concerns of the reviewers on the lack of related work and comparison of previous works, I recommend the paper as a poster presentation.